# Photolipid excitation triggers depolarizing optocapacitive currents and action potentials

Carlos A. Z. Bassetto Jr [1,5], Juergen Pfeffermann [2,5], Rohit Yadav [2,5], Simon Strassgschwandtner[2], Toma Glasnov [3], Francisco Bezanilla [1,4] ✉ & Peter Pohl [2] ✉

Optically-induced changes in membrane capacitance may regulate neuronal activity without requiring genetic modifications. Previously, they mainly relied on sudden temperature jumps due to light absorption by membrane-associated nanomaterials or water. Yet, nanomaterial targeting or the required high infrared light intensities obstruct broad applicability. Now, we propose a very versatile approach: photolipids (azobenzene-containing diacylglycerols) mediate light-triggered cellular de- or hyperpolarization. As planar bilayer experiments show, the respective currents emerge from millisecond-timescale changes in bilayer capacitance. UV light changes photolipid conformation, which awards embedding plasma membranes with increased capacitance and evokes depolarizing currents. They open voltage-gated sodium channels in cells, generating action potentials. Blue light reduces the area per photolipid, decreasing membrane capacitance and eliciting hyperpolarization. If present, mechanosensitive channels respond to the increased mechanical membrane tension, generating large depolarizing currents that elicit action potentials. Membrane self-insertion of administered photolipids and focused illumination allows cell excitation with high spatiotemporal control.

Regulating neuronal activity by light has sparked scientists' interest for several decades[1]. By allowing for tight spatiotemporal regulation, light-based approaches—notably optogenetics—aid in deciphering neuronal networks, supplementing traditional electrical techniques[2,3]. Optogenetic approaches frequently rely on light-gated cation-conducting channels, e.g., channelrhodopsins[4]. Exposing a channelrhodopsin-harboring neuron to light increases membrane cation permeability which causes depolarization and, eventually, neuronal spiking[5]. Effectively, this technique allows for millisecond-timescale and cell-type-specific optical control of neurons[6]. Yet, given that mammalian neurons do not generally express light-gated cation channels, genetic material encoding

these proteins has to be transfected and subsequently expressed by the target cells[7].

Photothermal optocapacitive approaches circumvent the necessity for introducing exogenous genetic material[8–10]. These techniques are based on a rapid light-evoked increase in cellular membrane temperature, leading to a fast rise in bilayer capacitance that causes a depolarizing optocapacitive current. Mechanistically, heating lipid bilayers causes them to expand in area and thin, entailing the required increase in capacitance[11,12]. The rate of change of capacitance determines action potential (AP)-generating effectiveness and critically depends on the rate of temperature change[10,13]. Membrane-associated extrinsic nanomaterials that convert visible light into heat[9], and high-energy pulses of

[1]Department of Biochemistry and Molecular Biology, The University of Chicago, Chicago, IL 60637, USA. [2]Institute of Biophysics, Johannes Kepler University Linz, Gruberstraße 40, 4020 Linz, Austria. [3]Institute of Chemistry, Karl-Franzens-University, Graz, Austria. [4]Centro Interdisciplinario de Neurociencia de Valparaíso, Facultad de Ciencias, Universidad de Valparaíso, Valparaíso, Chile. [5]These authors contributed equally: Carlos A. Z. Bassetto Jr, Juergen Pfeffermann, Rohit Yadav. ✉e-mail: fbezanilla@uchicago.edu; peter.pohl@jku.at

infrared light absorbed by water in the membrane vicinity[8] have achieved a sufficiently fast increase in membrane temperature for photothermal optocapacitive depolarization and consequent AP generation.

Photosensitive molecules represent yet another approach for regulating neuronal activity by light[14]. They can be classified by the mechanism of interaction with their target—non-covalent or covalent binding—as well as by the reversibility of their action[14]. For example, photoswitchable molecules covalently bound to glutamate receptors served to evoke or terminate trains of action potentials[15]. Non-covalently bound light-sensitive diacylglycerols with two azobenzene-containing acyl chains (OptoDArG) were used to regulate cation permeation and membrane potential by opening TRPC3 channels[16]. Albeit such photolipids may act according to a lock-and-key principle, as in the aforementioned examples, by partitioning into biological membranes and changing their molecular structure upon photoisomerization, they affect lipid bilayer material properties such as thickness, surface area, bending rigidity, compressibility and the propensity towards lipid domain formation[17–22]. It is well established that these properties may contribute to the gating of mechanosensitive and mechanically-modulated ion channels[23–26]. Photoisomerization-induced changes in bilayer material properties and tension may alter channel open probability or dynamics of membrane proteins[19,27,28].

Recently, neuronal excitation was achieved using the membrane-partitioning azobenzene-containing photoswitchable compound Ziapin2[29]. The photoisomerization of membrane-embedded trans-Ziapin2 by blue light led to a rapid decrease in membrane capacitance due to a change in Ziapin2's oligomeric state[29]. Conceivably, this drop produced a hyperpolarizing capacitive current. Yet, conductances of other origins also contributed as the observed hyperpolarizing current (i) did not linearly depend on voltage and (ii) persisted nearly unaltered for ≈250 ms even though the capacitance change took no longer than ≈20 ms[30]. Subsequently, delayed depolarization ensued by an elusive mechanism. Being unaware of the channels that might have facilitated the depolarizing current, the authors speculated that Ziapin2 caused a spontaneous fast capacitance increase ($t_{1/2} \approx 0.2$ s). However, Ziapin2's unprompted return to its cis conformation in DMSO takes orders of magnitude longer ($t_{1/2} = 108$ s)[30].

Here, we use minimal systems to identify the molecular mechanisms underlying photolipid-induced hyperpolarization and depolarization. We first use planar lipid bilayers to observe UV light-triggered depolarizing currents and subsequently exploit them to trigger APs by photolipid photoisomerization in sodium channel (Na$_V$1.3)-expressing HEK cells. We also clarify that mechanosensitive channels present in HEK cells may facilitate the depolarizing currents required for AP generation upon blue light-triggered hyperpolarization. We base this study on the photolipid OptoDArG (Fig. 1a), as diacylglycerols are (i) ubiquitous in biological systems, (ii) known to alter the mechanical properties of lipid bilayers upon photoswitching if equipped with azobenzene moieties[19], and (iii) able to spontaneously flip-flop across membranes which ensures their presence also in the inner leaflet of plasma membranes when added from the outside medium.

## Results and discussion

### Photolipid excitation elicits optocapacitive currents

Differences in molecular structure between OptoDArG's photoisomers affect the geometry of bilayers containing them: the extended cis-OptoDArG conformation awards bilayers with increased surface area, $A$, and reduced thickness of the membrane's hydrophobic core, $d_{hc}$, whilst its trans isomer requires less surface area per molecule and increases $d_{hc}$ (Fig. 1a)[19]. Consequently, photoisomerization of membrane-embedded OptoDArG changes lipid bilayer capacitance, $C$:

$$C = \varepsilon \times \frac{A}{d_{hc}} \qquad (1)$$

where $\varepsilon$ denotes absolute permittivity which we assume to be constant. Notably, the difference in the shape of the two monomeric OptoDArG photoisomers drives the changes in $d_{hc}$. In contrast, the recently reported Ziapin2 decreases $d_{hc}$ due to spontaneous dimer formation and increases $d_{hc}$ upon blue light-driven dimer dissociation[29]. For achieving optocapacitive modulation of the membrane potential, switching $C$ within tens of seconds to minutes as in our earlier study[19] would not suffice. As reflected by Eq. 2, the rate of change of $C$, d$C$/d$t$, determines the AP-generating effectiveness by the optocapacitive approach[10,13], i.e., the amplitude of the optocapacitive current, $I_{cap}$:

$$I_{cap} = \frac{dC}{dt}(V - V_s) + \frac{d(V - V_s)}{dt}C \qquad (2)$$

where $V$ and $V_s$ denote transmembrane and membrane surface potential differences[10,11,13]. Under voltage-clamp conditions d$V$/d$t$ = 0, and Eq. 2 simplifies to:

$$I_{cap} = (V - V_s) \times \frac{dC}{dt} \qquad (3)$$

Equation 3 assumes that $V_s$ remains unaltered during photoswitching. The condition is undoubtedly fulfilled in a planar lipid bilayer (PLB) with symmetrical leaflets, since $V_s \approx 0$ mV. For the plasma membrane of *E. coli* with $V_s = -18$ mV[31], a 5% change in the area per lipid would alter $V_s$ by no more than 1 mV, which is negligible.

To evoke $I_{cap}$ by photolipid photoisomerization, we folded horizontal solvent-depleted PLBs (diameter typically 60–80 μm) from 80 m% *E. coli* polar lipid extract (Avanti Polar Lipids) and 20 m% OptoDArG and placed them within working distance of the 40× magnification objective of an inverted widefield fluorescence microscope (Fig. 1b). Equation 3 predicts $I_{cap}$ in the range of tens to hundreds of picoampere for $V = 100$ mV and a 2.5 pF change in $C$ that occurs within a few milliseconds. The required time depends on the rate of isomerization (see below).

Before exposure to blue laser light (≈30 mW, ≈58 μm 1/e²-diameter, 488 nm), a UV laser diode (375 nm) switched OptoDArG into a cis-enriched photostationary state. Blue light illumination triggered positive $I_{cap}$ when we clamped the PLB at negative $V$, and negative $I_{cap}$ for positive $V$ (Fig. 1c). Since photoisomerization from cis- to trans-OptoDArG by blue light reduces $C$, this result is in line with Eq. 3. Further, this observation rules out a photothermal optocapacitive mechanism induced by blue light because heating thins the membrane, i.e., mandates d$C$/d$t > 0$ which would generate oppositely-directed $I_{cap}$[8,10,11]. Subsequent rapid switching of the PLB into a cis-enriched photostationary state using UV laser light (≈30 mW incident at a diameter of roughly 150–250 μm) generated positive $I_{cap}$ at positive $V$ (Fig. 1d). Again, this is consistent with Eq. 3 since cis-OptoDArG increases $C$ and, thus, d$C$/d$t$ is positive.

To determine the factors governing $I_{cap}$, we begin by expressing the time course of $C$ upon blue, $C_{blue}$, and upon UV light exposure, $C_{UV}$, as follows:

$$C_{blue}(t) = \Delta C_{blue}(t) + C_0 \text{ and } C_{UV}(t) = \Delta C_{UV}(t) + C_0 \qquad (4)$$

where $C_0$ is $C$ at the onset of light exposure, $\Delta C_{blue}$ the decrement in $C$ upon blue light exposure, and $\Delta C_{UV}$ the increment in $C$ upon UV light exposure. Changes in $C$ upon photoisomerization emerge from the structural transition of many individual photolipids. As part of the bilayer capacitor, each cis and trans photolipid contributes to $C$, and we refer to the individual contributions as $c_c$ and $c_t$; conceivably, $c_c > c_t$. Consequently, $\Delta C_{blue}$ and $\Delta C_{UV}$ depend on the number of cis- and trans-OptoDArG molecules in the bilayer $n_c$ and $n_t$:

$$\Delta C_{blue}(t) = (n_c(t) - n_{c,0}) \times \Delta c \text{ and } \Delta C_{UV}(t) = (n_{t,0} - n_t(t)) \times \Delta c \qquad (5)$$

whereby $\Delta c = c_c - c_t$. $n_{c,0}$ and $n_{t,0}$ denote the number of cis and trans photolipids at time $t = 0$ s. $n_c$ and $n_t$ decay depending on blue, $I_{blue}$, and UV light irradiance, $I_{UV}$, as:

$$n_c(t) = n_{c,0}\exp(-k_{ct}I_{blue}t) \text{ and } n_t(t) = n_{t,0}\exp(-k_{tc}I_{UV}t) \quad (6)$$

where $k_{ct}$ and $k_{tc}$ (in $m^2W^{-1}s^{-1}$) denote rate constants of OptoDArG photoisomerization from cis to trans and trans to cis. Though of the same order of magnitude, $k_{ct}$ and $k_{tc}$ adopt different values[32]. They may be considered to be independent of power[32]. Also, Eq. 6 considers $k_{tc}$ upon blue and $k_{ct}$ upon UV light exposure to be zero, which, albeit a good approximation, is quantitatively incorrect due to overlaps in

photoisomer absorption[32,33]. Equation 6 also assumes a negligibly small contribution of thermal relaxation of the metastable cis photoisomer at the millisecond timescale[22]. From Eqs. 3 and 6 we find:

$$\frac{I_{cap,blue}}{V} = \frac{dC_{blue}(t)}{dt} = -\Delta c \times n_{c,0}k_{ct}I_{blue}\exp(-k_{ct}I_{blue}t),$$
$$\frac{I_{cap,UV}}{V} = \frac{dC_{UV}(t)}{dt} = \Delta c \times n_{t,0}k_{tc}I_{UV}\exp(-k_{tc}I_{UV}t) \quad (7)$$

where $I_{cap,blue}$ and $I_{cap,UV}$ denote optocapacitive currents evoked upon blue and UV light exposure.

For small sizes of the solvent torus anchoring the PLB and millisecond intervals, we may assume that $n_c + n_t$ is constant, i.e., lipid

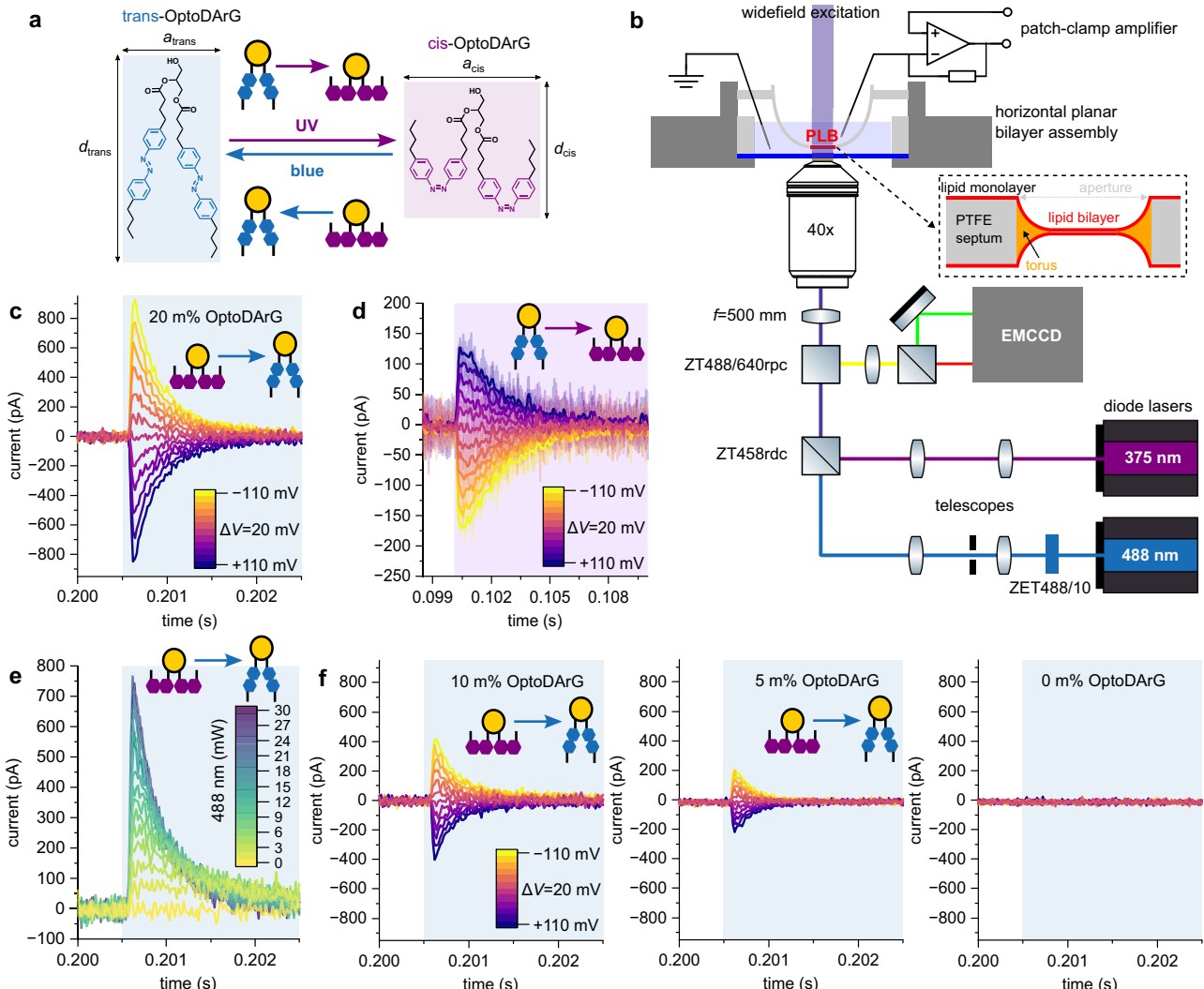

**Fig. 1 | Generation of hyper- and depolarizing optocapacitive currents in PLBs.**
**a** Rationale for capacitance changes upon OptoDArG photoisomerization: cis-OptoDArG is a broader ($a_{cis} > a_{trans}$) and shorter ($d_{cis} < d_{trans}$) molecule than trans-OptoDArG. Upon incorporation into bilayers, these differences in molecular structure alter bilayer geometry upon photoisomerization: UV light favors cis-OptoDArG, which increases bilayer surface area and reduces thickness. Blue light achieves opposite effects. According to Eq. 1, these geometrical alterations translate into differences in capacitance. **b** A schematic representation of the horizontal PLB setup. For details on the setup, the formation and structure of PLBs, see Methods. **c** Optocapacitive currents were generated upon rapid photoisomerization of membrane-embedded cis-OptoDArG by blue light (blue background). The PLB was clamped at voltages ranging from −110 mV to +110 mV with $\Delta V = 20$ mV, as indicated by the color legend. Current traces obtained at different $V$ are overlaid. The laser was kept on for 50 ms to ensure quantitative

switching whereby the evoked currents decayed to electrical noise level within milliseconds. **d** Recorded as **c** but the PLB was in its trans photostationary state prior to UV light exposure (purple background). The raw currents are overlaid with smoothed curves for clarity. In comparison to **c**, maximum current amplitude and rate of current decay are apparently reduced; as predicted by Eq. 7, this is a consequence of the lower irradiance of UV relative to blue light (see Methods). **e** Overlay of optocapacitive currents recorded at $V = -110$ mV and different blue light intensity (the numbers in the color legend give power at the sample stage in mW). **f** To demonstrate the dependence of $I_{cap}$ on the fraction of OptoDArG in the PLB-forming lipid mixture (see Eq. 7), the latter was reduced from 20 m% (as in **c**–**e**) to 10 m% and 5 m%. At 0 m%, blue light exposure evokes no currents. The recordings were done as in **c**. The experimental data is from $N = 10$ independently prepared experiments, each comprising multiple technical replicates.

trafficking between the bilayer and the PLB's lipid reservoirs (torus and monolayers) is negligible. Then Eq. 7 predicts an exponential decay of $I_{cap}$—which we tentatively observe in Fig. 1c, d. Indeed, we find that the power-dependent $I_{cap,blue}$ in Fig. 1e are well-grasped by simple exponential decay fits (Supplementary Fig. 1a). The obtained mono-exponential rate constants $k_{blue} = k_{ct}I_{blue}$ are linear with irradiance up to ≈750 Wcm$^{-2}$ blue light with $1/k_{blue}$ in the millisecond range. $k_{blue}$ levels off abruptly at >750 Wcm$^{-2}$, conceivably due to instrumental (filtering) and fitting limitations (Supplementary Fig. 1b). $1/k_{blue}$ is orders of magnitude larger than the time (ps) required for switching individual azobenzene moieties[34] because the absorption probability is limiting. A linear model fit reveals a slope of $k_{ct} = 3.85 \times 10^{-3}$ cm$^2$mW$^{-1}$s$^{-1}$ ($R^2 > 0.99$). We note that our blue illumination profile was not flat-top but Gaussian, so we estimate irradiance by calculating the area from the $1/e^2$-diameter of the illumination profile (≈ 58 μm). $k_{ct}$ is reasonably close to the rate of photoisomerization of azobenzene-containing surfactants of $3 - 4 \times 10^{-3}$ cm$^2$mW$^{-1}$s$^{-1}$ at 490 nm[32]. This agreement indicates that the observed $I_{cap}$ values are an immediate consequence of changes in $C$ due to photolipid photoisomerization. As also predicted by the model, $k_{blue}$ and $k_{UV}$ are independent of $V$ (Supplementary Fig. 2).

In contrast to the prediction of Eq. 7 and the experimental results in PLBs shown in Supplementary Fig. 3a, a previous report claiming to have observed optocapacitive currents[29] displayed light-triggered currents that persisted nearly unaltered as long as light exposure lasted[30]. The molecular origin of these persisting currents remained elusive. Perhaps the dimerization propensity of Ziapin2, the molecule with azobenzene moieties replacing photolipids in the latter study, is responsible for it. Alternatively, the activation of membrane ion channels may provide an explanation. It is important to note that in the absence of OptoDArG no optocapacitive currents are observed, further demonstrating that the effects observed in PLBs containing OptoDArG are not light-induced thermal effects (Supplementary Fig. 3b).

Equation 7 proposes two straightforward strategies to enhance $I_{cap,blue}$, and $I_{cap,UV}$:

a. Increasing $I_{blue}$ or $I_{UV}$ ought to increase d$C$/d$t$. For demonstration, we clamped a PLB at $V = -110$ mV and repeatedly exposed it to blue light of increasing intensity (Fig. 1e).
b. As demonstrated in Fig. 1c and f, decreasing OptoDArG concentration, i.e., $n_{c,0}$ and $n_{t,0}$ in the PLB-forming lipid mixture from 20 m% to 5 m% reduces initial optocapacitive current amplitude.

## Light-induced capacitance changes generate $I_{cap}$

Consecutive 50 ms blue (blue background) and UV (purple background) light exposures induced fast changes in $C$ of PLBs (Fig. 2a, b). Zooming into the first milliseconds of the exposure times (Fig. 2c, d) reveals exponentially decaying capacitance traces. Differentiating the latter (Eq. 3) numerically by calculating difference quotients and multiplying the obtained derivative traces by $V = \pm 110$ mV yielded current traces. Overlaying these calculated traces with $I_{cap}$ (obtained under voltage-clamp at $V = \pm 110$ mV) showed a reasonable match with the recorded current (Fig. 2e, f). The small residual deviations may be due to (i) the time delay between $I_{cap}$ and $C$ recordings and (ii) limitations set by the rate with which we can measure membrane capacitance. The electrodes and the deployed software lock-in amplifier (HEKA) limited us to 5 kHz sine frequency of the probing signal. Ideally, we would have aimed for a rate closer to 20–50 kHz.

Vice versa, we note that the photoisomerization-induced $\Delta C$ can be found by integrating $I_{cap}$ (Eq. 3):

$$Q_{cap} = \int_{t_0}^{t_f} I_{cap}\,\mathrm{d}t = (V - V_s) \times \int_{t_0}^{t_f} \frac{\mathrm{d}C}{\mathrm{d}t}\,\mathrm{d}t$$
$$= (V - V_s) \times (C(t_f) - C(t_0)) = (V - V_s) \times \Delta C \qquad (8)$$

where $Q_{cap}$ denotes the number of capacitive charges displaced upon photoisomerization, $t_0$ the starting time of light exposure, and $t_f - t_0$ the integration time. To demonstrate this approach, we numerically integrated $I_{cap}$ in Fig. 1c, d deploying an integration time of 3 ms and 10 ms, respectively. Figure 2g, h show the resulting values for $Q_{cap}$ plotted against $V$; according to Eq. 8, the slope of the indicated linear fits equals $\Delta C$. We find that the thus obtained values for $\Delta C$ (−3.28 pF and +3.49 pF) underestimate those obtained from direct capacitance recordings (−3.57 pF (Fig. 2c) and +4.54 pF (Fig. 2d)) by ≈10–25%. These deviations partly originate from finite integration times and analog filtering of the current traces at 10 kHz.

Upon closer inspection of the capacitance traces in Fig. 2a, b, we find that $C$ drops to a minimum value within milliseconds (Fig. 2c) following blue light exposure (at 1.15 s) and then relaxes quickly; we refer to the change in $C$ after relaxation relative to the initial level pre-exposure as $\Delta\Delta C$ (indicated in Fig. 2a, b). Consistent with the 3 ms integration time used to determine $\Delta C$ from $I_{cap}$ (Eq. 8), $\Delta C$ corresponds to the drop in $C$ within the first 3 ms following light exposure, as indicated in Fig. 2a, b. In the presence of a small torus, we observe that this fast relaxation in $C$ (between 1.2 and 1.3 s in Fig. 2a, indicated by the gray arrow) amounts to <15% of the amplitude of the drop, whilst it can reach >30% in the presence of a larger torus (Fig. 2b; for details on how the large torus was enforced, see Methods). Our interpretation of the results (Fig. 2a, b) is the following: Blue light exposure quantitatively decreases PLB area within a few ms, leading to a pulling force of the PLB on the torus and hence the build-up of membrane tension—$C$ decreases to its minimal value (Fig. 2c). The force originates from area differences between cis and trans photolipids, as is explored further in the following section. Then, bilayer material is drawn out of the torus, causing at least a partial relaxation of membrane tension and the observed increase in $C$. In the presence of a larger solvent-containing torus, more material can be pulled out of the torus and hence, the amplitude of fast relaxation (i.e., $\Delta\Delta C - \Delta C$) is larger (Fig. 2b).

Interestingly, relaxation in $C$ also follows the photolipid-mediated increase in $C$ upon UV light exposure. Conceivably, it signifies that the constitutive tension of PLBs flattens photo-induced undulations or PLB bulgings that appear with the increment in membrane area upon membrane thinning. Importantly, these torus-specific relaxations do not represent an intrinsic feature of the photolipid. These minor drawbacks of the experimental model system notwithstanding, we find a general agreement between current and capacitance recordings. Applications of Eq. 3, Eq. 7 and Eq. 8 provide strong evidence for the optocapacitive mechanism of rapid photolipid photoisomerization.

## The transition to trans-OptoDArG generates membrane tension

Rapid photoisomerization from cis- to trans-OptoDArG generates membrane tension, as schematically depicted in Fig. 2i, j. First, exposure to blue light reduces area per photolipid ($a_{trans} < a_{cis}$). Second, the narrower photolipids exert a pulling force on the neighboring lipids and thus stretch them temporarily (illustrated by the green springs). Equivalently, one may consider that the reduction in photolipid area leads to a transient reduction in bilayer-internal lateral pressure due to less steric hindrance[27]. The result is the same: tension components temporarily outweigh pressure components in the bilayer's lateral pressure profile, and, as a result, membrane tension emerges[35]. Notably, albeit the underlying molecular mechanisms may be distinct—i.e., other photolipids may alter their oligomeric state or partition in/out of the membrane upon switching—changes in the area of lipid bilayers[19,36,37] and cellular membranes[29,38] upon photoisomerization of membrane-targeted azobenzene-containing molecules have been reported by multiple groups.

As a back-of-the-envelope calculation shows, the achievable tension, $\tau$, is relevant for mechanosensitive proteins. For example, mechanosensitive channels like MscL open at $\tau \approx 10$ mN/m[39]. Such $\tau$ value translates for membranes with a typical stretching modulus, $K$, of

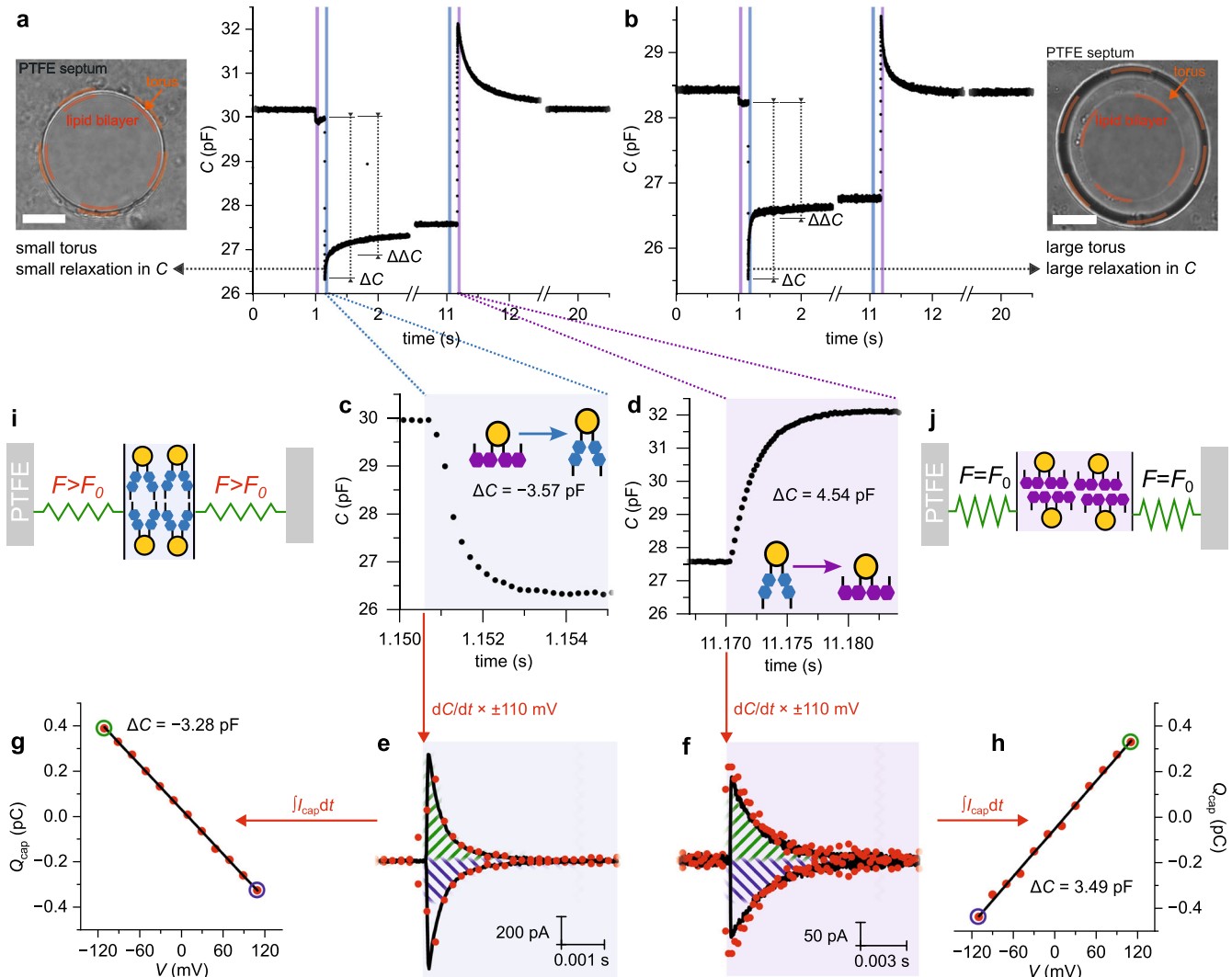

**Fig. 2 | Light-induced capacitance changes generate $I_{cap}$. a, b** Time trace of PLB capacitance (black line; an average of 5 consecutively-recorded traces) indicating the 50 ms exposure intervals to blue light (blue background) and UV light (purple background). UV illumination immediately prior to blue light exposure at 1.15 s and blue light illumination immediately prior to UV light exposure at 11.17 s ensures a cis- and trans-enriched photostationary state, respectively. In **a**, the torus of the solvent-depleted PLB was small (image on the left). In **b**, $C$ was recorded in the presence of a large solvent torus, as apparent in the corresponding PLB image on the right. Note the increased relaxation amplitude following blue light exposure relative to **a**. Scale bar is 25 μm in both images. **c, d** $C$ changes exponentially as zoom-ins into the (**c**) blue (blue background) and (**d**) UV light (purple background) illumination intervals shown in **a** demonstrate. **e, f** The graphs below the recordings of $C$ show $I_{cap}$ with (**e**) blue and (**f**) UV light exposure recorded at $V = \pm 110$ mV (black lines; each averaged from 20 consecutively-recorded current traces). The overlaid red points are calculated from the respective above capacitance trace by numerical differentiation and subsequent multiplication with $\pm 110$ mV, as indicated by the arrows. **g, h** The graphs show $Q_{cap}$ obtained upon integration of $I_{cap}$ in Fig. 1c (**g**) and Fig. 1d (**h**) plotted over $V$; acc. Equation 8, the slope of a linear fit (black line) corresponds to $\Delta C$ upon photoisomerization ($R^2 > 0.99$). The green and blue circles correspond to the areas marked with the same color in **e** and **f. i, j,** Schematics of how photoisomerization from cis- to trans-OptoDArG generates tension within the membrane. The grey bars labeled PTFE symbolize the septum aperture within which the PLB is mounted, and green springs refer to the bilayer lipids other than OptoDArG. Upon blue light exposure, the area per photolipid decreases ($a_{trans} < a_{cis}$) which, as the PLB is anchored laterally, causes stretching of the remaining lipids in the bilayer. The loading of the springs symbolizes tension within the PLB. The experimental data is from $N = 7$ independently prepared experiments, each comprising $n > 10$ technical replicates.

250 mN/m into an area increment of 4% ($\alpha = 0.04$), since $\tau = K \times \alpha$. Thus, the energy, $E_r$, required to stretch a membrane of area $A_M$ is equal to:

$$\frac{E_r}{A_M} = \frac{1}{2}\tau \times \alpha = 0.2\,\frac{mN}{m}$$

The work performed by isomerization of a single azobenzene switch has been estimated at $E_p = 4.5 \times 10^{-20}$ J[40]. Assuming an Opto-DArG concentration of 10 mass-% and an average area per lipid of 0.7 nm², we find one photolipid in an area $A_L = 7$ nm². Since the

photolipid contains two azobenzene switches we can write:

$$\frac{E_p}{A_L} = \frac{2 \times 4.5 \times 10^{-20}\,J}{7\,nm^2} = 13\,\frac{mN}{m}$$

A comparison of the two above equations reveals that:

$$\frac{E_p}{A_L} \gg \frac{E_r}{A_M}$$

Thus, it is safe to conclude that photolipid switching may provide the energy required to open mechanosensitive channels, even if

(i) <100% of the photolipids change their conformation, (ii) the deformation of other lipids and flattening of membrane undulations requires some extra energy.

Eventually, the photo-induced tension will relax because additional lipids from the torus may be pulled into the PLB. The situation may be similar in cells as membrane invaginations may also provide a lipid reservoir, as previously observed with osmotically challenged cells[41]. We demonstrate the photoactivation of mechanosensitive channels below to provide experimental evidence for photo-induced tension in the cellular membrane.

### Photolipid-evoked hyper- and depolarizing currents in cells

OptoDArG self-inserts into the exofacial leaflet of the plasma membrane when dissolved in DMSO and administered to the aqueous solution. Since diacylglycerols may rapidly flip-flop in model and biological membranes[42], OptoDArG also rapidly reaches the cytoplasmic leaflet. Equilibration is slower only in sphingolipid-enriched membranes[43]. We observed that cells remained healthy after 1 h of labeling and patch-clamp experiments could be performed for hours after. The azobenzene photoswitch generally combines good photophysical and pharmacokinetic properties with low phototoxicity, e.g., low singlet oxygen generation[14]. In cellular systems under experimental conditions, exposing HuH7 cells to lipid vesicles comprising the azobenzene-containing photoswitchable phospholipid azo-PC does not impact cellular viability in the absence of irradiation for 24 h[44]. The cellular tolerance to photolipids is further supported by a recent preprint study which reports that HeLa cells can incorporate FAAzo4, a synthetic azobenzene-containing fatty acid, into glycerophospholipids, thereby generating photoswitchable lipids without compromising cellular viability over the course of 24 h[45]. For the cell experiments, we used a setup optimized for whole-cell patch-clamp experiments (Fig. 3a). The intensity of the Ti: Sapphire laser (367 nm after the second harmonic generator) was regulated by a polarizer. The pulse duration was controlled by a shutter that allowed the generation of millisecond pulses with a sharp rise time (<20 μs). An in-house designed and built uncoated fused silica objective optimized the intensity of UV light reaching the sample stage.

As in PLBs, we found that exposing the cells (Fig. 3b) to millisecond UV and blue light pulses (445 nm) generated hyper- and depolarizing currents, respectively (Fig. 3c). In accordance with Eq. 7 and in line with our PLB recordings (e.g., Fig. 1e), the optocapacitive currents in Fig. 3c decay exponentially. This is in stark contrast to a thermal optocapacitive mechanism, where exposure to light causes an increased current until thermal equilibrium is reached, which typically requires much longer than the millisecond pulses used here[8]. Further, no optocapacitive current was observed when cells were not labeled with OptoDArG (Supplementary Fig. 4a, b), indicating that the currents shown in Fig. 3c were not thermally evoked. The conclusion is based on the observations that membrane capacitance is a perfect indicator of temperature changes[8,46] and $C$ did not change upon laser illumination in the absence of the photolipid. It is important to note that when the blue laser power exceeded 200 mW, it was possible to observe thermal optocapacitive effects (Supplementary Fig. 4c). Therefore, we never exceeded 160 mW when conducting experiments using the blue laser. Whilst these control experiments do not strictly rule out the possibility that the interaction of light and OptoDArG leads to heating, the symmetry of the observed light-evoked capacitance changes upon UV and blue light illumination does (for details see further below).

Subsequently, we sought to gain a rough estimate for the amount of membrane-embedded OptoDArG. We did so by capacitance measurements, measuring the time integral of the transient current for voltage steps (Fig. 3d). A measurement of $C$ in the cis photostationary state, $C_{cis}$, and the trans photostationary state, $C_{trans}$, is shown in Fig. 3e. The result of measurements on 10 cells is shown in the left

panel of Fig. 3f. Subsequent calculation of the percentage change in $C$ results in a distribution of values (middle panel in Fig. 3f), frequently around 5%. Conceivably, the incorporation of OptoDArG into cells was less homogenous than into PLBs as can be seen from the spread of percentage change in $C$ in Fig. 3f. Nevertheless, PLBs doped with 10 m% OptoDArG yielded similar changes in capacitance, suggesting that the cellular membranes may also have contained up to 10 m% OptoDArG (Figs. 2 and 3).

### OptoDArG triggers photo-induced APs

As entailed by Eq. 2, optocapacitive currents modulate the membrane potential. To explore this, we conducted whole-cell recordings in current-clamp mode using HEK293 permanently transfected with $Na_V1.3$. To initiate an AP, only $Na^+$ currents carried by $Na_V$ channels are necessary[47], which will depolarize the cell membrane. In our case, we exploited endogenous HEK293 leak channels and leakage currents through the seal to repolarize the cell membrane. Therefore, the HEK293 $Na_V1.3$ can be regarded as an artificial neuron regarding the generation of an AP. In HEK293 $Na_V1.3$, exposure to UV light generated rapid optocapacitive depolarization (Fig. 4a), in agreement with the observation of depolarizing capacitive currents under voltage-clamp (Fig. 3c). Depolarization subsequently triggered the opening of $Na_V1.3$ channels which generated an action potential (Fig. 4a). When there was no OptoDArG incorporated in the cells, there was neither depolarization (Supplementary Fig. 4d) nor optocapacitive current generated upon UV light exposure (Supplementary Fig. 4a,b), which refutes UV light-triggered thermal effects.

Exposure to blue light evoked hyperpolarization, which was followed by depolarization and subsequent AP generation (Fig. 4b). Since the transition to trans-OptoDArG is associated with the generation of membrane tension (Fig. 2i), we added $Gd^{3+}$, a broad inhibitor of mechanosensitive channels[48,49]. When repeating the experiment in the presence of 5 μM $Gd^{3+}$, the large depolarizing currents were inhibited and photostimulation with blue light did no longer evoke APs (Fig. 4c). This differential response to blue light exposure between the absence (Fig. 4b) and presence of $Gd^{3+}$ (Fig. 4c) was not due to $Na_V1.3$ being blocked by 5 μM $Gd^{3+}$ (Supplementary Fig. 4e); also, AP generation upon UV light exposure did not depend on the presence (Fig. 4a) or absence of 5 μM $Gd^{3+}$ (Fig. 4d). We conclude that the rapid transition to trans-OptoDArG mediated by intense blue light triggered a $Gd^{3+}$-sensitive depolarizing current, potentially facilitated by endogenous mechanosensitive channels[50].

The success rate for AP generation by UV in the absence of $Gd^{3+}$ was 33.3% and in the presence of $Gd^{3+}$ was 13.3% (see Supplementary Tables 1 and 2). We believe that the moderate success rate for the condition without $Gd^{3+}$ is due to the variability in cells. The rate of success is smaller in the presence of $Gd^{3+}$ due to the fact that $Gd^{3+}$ might be blocking $Na^+$ channels and/or shifting their voltage-dependence to the right (to more positive voltages) by changing the surface potential; this makes it harder for a given depolarization to induce an AP. When we compare the amount of depolarization induced by UV light-driven photoisomerization of OptoDArG under current-clamp conditions (column "$\Delta V$ (mV) by UV" in Supplementary Tables 1 and 2), we can observe that it varies from as little as 4–5 mV to up to almost 20 mV. This variability is expected considering the apparent variability in fractional capacitance changes shown in Fig. 3f. Additionally, there are instances where the UV-mediated depolarization was ≈7 mV and it was sufficient to elicit an AP, whereas for another cell it was ≈15 mV and no AP was generated. Again, this indicates that low success rates are due to the cells' condition prior to excitation rather than due to the photolipid technique's capability to generate depolarization. To elicit an AP in a cell many factors play a role. For example, the amount of sodium channels expressed, the current carried by them and the level of inactivation that they have prior to the depolarizing stimulus. Another factor is related to the input resistance

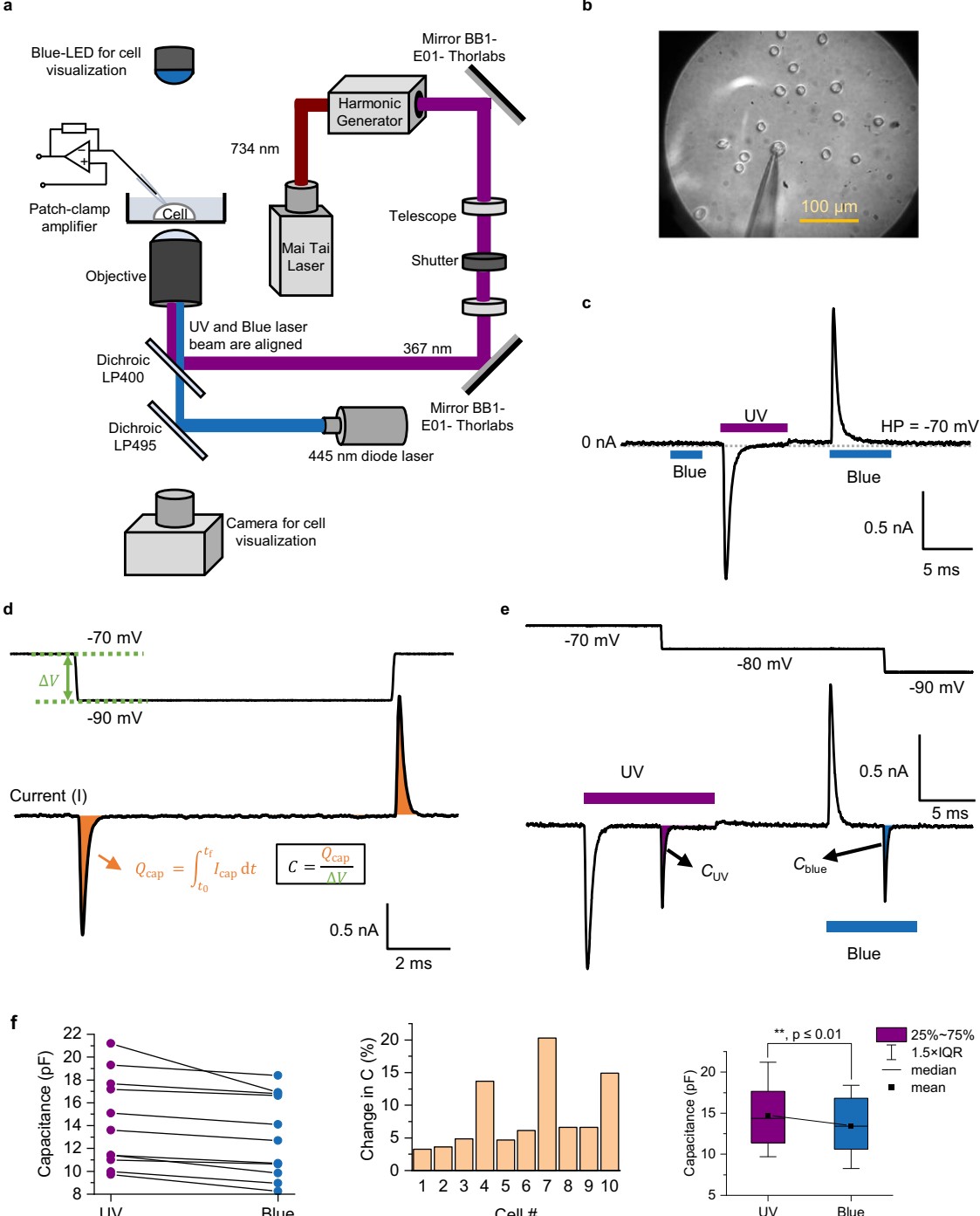

**Fig. 3 | OptoDArG photoisomerization generates hyper- and depolarizing capacitive currents in HEK293 expressing Na$_V$1.3. a** Schematic representation of the cell stimulation setup; details are provided in the Methods section. **b** HEK cells were labeled with 60 μM OptoDArG for 1 h (the labeling protocol is described in the Methods section). The image was taken through the in-house designed fused silica objective. **c** Whole-cell voltage-clamp recordings on HEK293 cells expressing Na$_V$1.3 labeled with 60 μM OptoDArG demonstrate the generation of depolarizing currents upon UV (60 mW) and hyperpolarizing currents upon blue light exposure (60 mW). The cell was held at −70 mV during the laser pulses; the duration of exposure is indicated by bars in the current trace. The current trace was recorded in the presence of 50 μM Gd$^{3+}$ added to the external solution (Methods section). **d** Method used for inferring changes in $C$ upon photoisomerization: Under voltage-clamp, applying a voltage step, $\Delta V$, to the membrane evokes a capacitive current. This current is given by Eq. 2 under the condition that d$C$/d$t$ = 0 and d$V_s$/d$t$ = 0. Integration of this expression analogous to Eq. 8 results in $Q_{cap} = C \times \Delta V$. Thus, $C$ is

inferred from a numerical integration of the capacitive current peak evoked upon a voltage step. **e** Optocapacitive currents were generated by UV and blue light exposure; after these had decayed, voltage steps queried the instantaneous value of $C$, as described in **d**. Thus, the voltage step following UV light reveals $C$ in the cis photostationary state, $C_{cis}$, and following blue light $C$ of the trans photostationary state, $C_{trans}$. To obtain absolute values for $C_{cis}$ and $C_{trans}$, slow capacitance compensation was off. As in **c**, the displayed current trace was recorded in the presence of 50 μM Gd$^{3+}$. **f** The left panel gives $C_{cis}$ and $C_{trans}$ determined as described under **e** for 10 different cells ($N$ = 10 biological replicates from fresh preparations). As observed in PLBs, consistently $C_{cis} > C_{trans}$. The center panel gives the corresponding percentages of change in $C$, ($C_{cis} - C_{trans}$)/$C_{cis} \times 100$. The right panel shows the result of a one-sided paired $t$-test (hypothesis: mean $C_{cis\,(UV)} > C_{trans\,(Blue)}$) on the above data, demonstrating that at $p \leq 0.01$ ($p = 0.00311$), $C_{cis}$ is different from (larger than) $C_{trans}$.

**a**   Action potential induced by OptoDArG 120 μM + 5 μM Gd³⁺

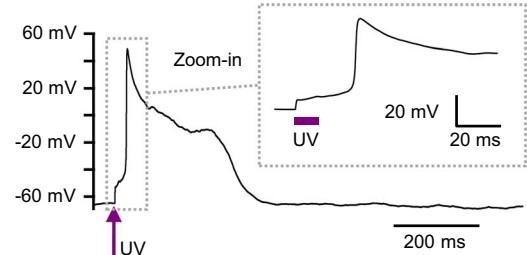

**b**   Action potential induced by mechanosensitive currents
OptoDArG 60 μM without Gd³⁺

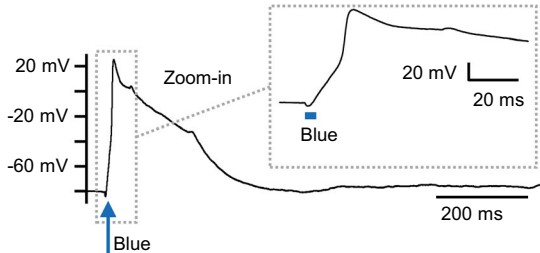

**c**   No action potential induced by OptoDArG 120 μM + 5 μM Gd³⁺

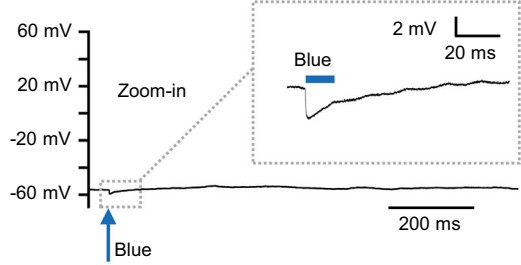

**d**   Action potential induced by OptoDArG 60 μM without Gd³⁺

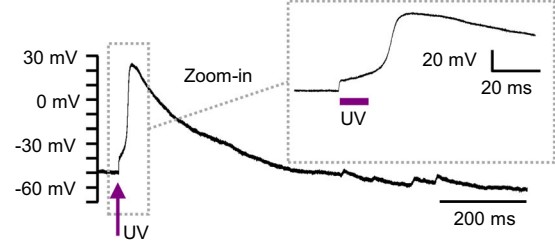

**e**   OptoDArG 60 μM + 5 μM Gd³⁺

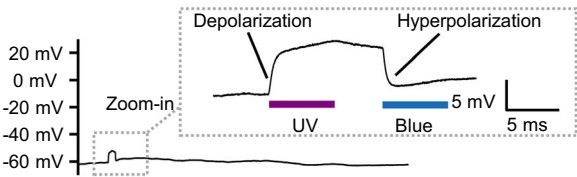

**f**   OptoDArG 60 μM without Gd³⁺

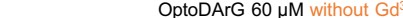
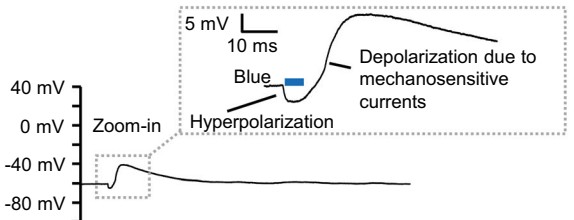

**Fig. 4 | Photo-induced APs.** The panels show current-clamp recordings of HEK293 Na$_V$1.3 under varying conditions. UV laser power was 120–121 mW; blue laser power was 60 mW. **a** AP elicited by OptoDArG-mediated optocapacitive depolarization upon UV light exposure recorded in the presence of 5 μM Gd³⁺; OptoDArG was 120 μM. The AP shown in **a** was recorded under conditions where the driving force for Na⁺ ions was augmented (see Methods). **b** In the absence of Gd³⁺, hyperpolarization upon blue light exposure is followed by depolarization and AP generation; OptoDArG was 60 μM. Depolarization is attributed to mechanosensitive currents activated by the transition from cis- to trans-OptoDArG (**f**). **c** In the presence of 5 μM Gd³⁺ and 120 μM OptoDArG, blue light causes hyperpolarization, and neither depolarization nor AP generation. **d** Same as **a**, but in a different cell, in the absence

of Gd³⁺ and with 60 μM OptoDArG. **e** OptoDArG-induced depolarization with UV followed by hyperpolarization with blue illumination. **f** Blue light-driven hyperpolarization induced by cis to trans photoisomerization of OptoDArG and subsequent depolarization induced by mechanosensitive currents. Note that in some cases the depolarization generated by the photolipid currents did not reach the threshold to generate an AP, as shown in (**e**). Similarly, in some cases, the depolarization induced by mechanosensitive currents did not reach the threshold to fire an AP, as shown in (**f**). Whilst this figure serves as a summary of our findings, Supplementary Tables 1 and 2 contain experimental details on the 52 cells assayed (freshly prepared biological replicates).

of the cell. In our hands, the cells expressing Na$_V$1.3 required a large depolarization to elicit an AP and sometimes even with a depolarization of 30–35 mV cells failed to fire (Supplementary Fig. 5), whereas in other cases 20 mV was enough to elicit an AP (column "Δ$V$ (mV) by injecting current" in Supplementary Tables 1 and 2), emphasizing the cellular variability. Therefore, we conclude that our low success rates for evoking APs by OptoDArG-mediated depolarization are due to cell variability rather than due to the photolipid optocapacitance technique.

Next, we studied the blue-light evoked capacitive and ionic currents under voltage-clamp conditions in cells and found further evidence for the contribution of mechanosensitive channels. Immediately following the hyperpolarizing optocapacitive current, another current emerges in the absence of Gd³⁺ (Fig. 5a). We find that this current is non-selective, reversing at around 0 mV (in sharp contrast to the optocapacitive current), its activation is not voltage-dependent, and it is sensitive to Gd³⁺. In the presence of 10 μM Gd³⁺ these ionic currents are subdued (Fig. 5b), as is blue light-triggered delayed depolarization (Fig. 4c). Most likely, they are facilitated by endogenously expressed

mechanosensitive channels present in HEK293. We also observed the light-evoked mechanosensitive currents in blank HEK293T cells (not expressing Na$_V$1.3), evidencing that the currents were not carried out by Na$_V$1.3 (Supplementary Fig. 6) and suggesting that they are from an endogenous channel. Since HEK293 cells endogenously express a myriad of different ion channels[51], the mechanosensitive channel responsible for the hyperpolarizing currents observed here remains yet to be identified.

Plotting the amplitude of the hyperpolarizing current vs holding voltage reveals a large positive reversal potential, $V_{rev}$, in the range of +230 mV (Fig. 5c). It is recorded in the presence of Gd³⁺ as determining the reversal potential is otherwise hampered by the current generated by mechanosensitive channels. The potential is too large to be attributed to the intrinsic difference in HEK cell surface potential alone. Yet, Gd³⁺ may have contributed as its one-sided adsorption alters this difference profoundly[8,11]. Figure 5d highlights the rapid ms-timescale changes in capacitance of OptoDArG-containing cellular membranes upon UV and blue light illumination that are critical to the optocapacitive method[10,13]. Coincidentally, Fig. 5d refutes underlying thermal

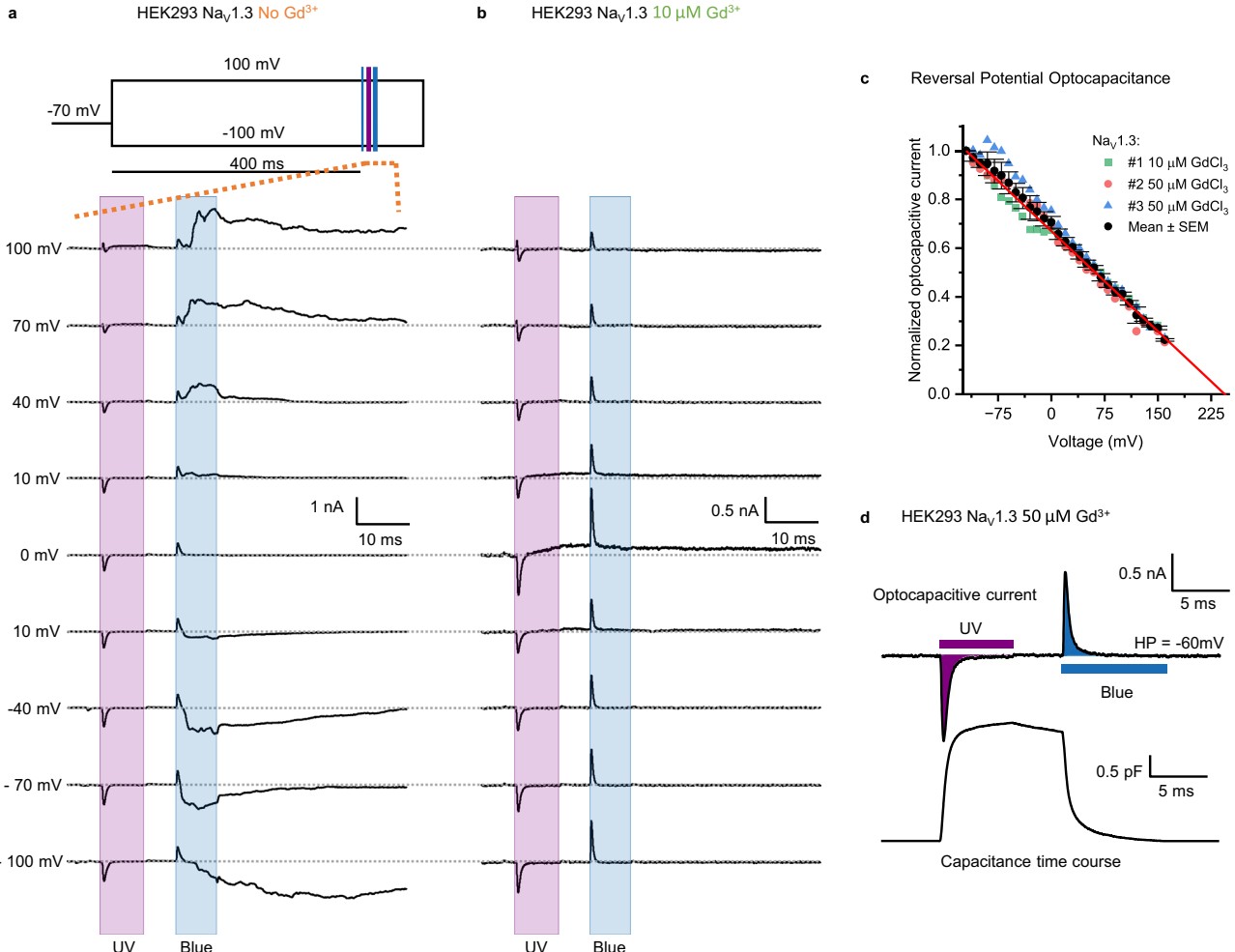

**Fig. 5 | Voltage-clamp study on Gd³⁺-sensitive currents evoked upon switching to trans-OptoDArG.** Panels **a** and **b** show voltage-clamp recordings of HEK293 Na$_V$1.3. UV laser power was 97–100 mW; blue laser power was 60 mW. The voltage protocol for **a** and **b** is illustrated in the inset of **a**; the shown period of the obtained current traces is indicated by orange dashed lines. We used three laser pulses: (1) a blue pulse (3 ms) to achieve the trans photostationary state; (2) a UV pulse (6.5 ms) to photoisomerize trans- to cis-OptoDArG which generates a depolarizing current; (3) another blue laser pulse (6 ms) rapidly achieves trans-OptoDArG which results in a hyperpolarizing current. Both, depolarizing and hyperpolarizing capacitive currents in **a** and **b** reverse at large positive $V$, qualitatively consistent with thermally-evoked optocapacitive currents in cells. **a** In the absence of Gd³⁺, the blue light-evoked optocapacitive current is followed by a current that reverses at around 0 mV. To avoid contamination of the mechanosensitive currents with the Na⁺ currents carried by Na$_V$1.3, we applied a 400 ms depolarizing pulse to ensure that all sodium channels were in the inactivated state before evoking the mechanosensitive current by the laser pulses. **b** In the presence of 10 μM Gd³⁺, the hyperpolarizing capacitive currents decay, and no additional currents are triggered. Cells in **a** and **b** are from different dishes. **c** Graph of normalized blue light-evoked hyperpolarizing current amplitude over clamped voltage; a linear fit reveals the large positive reversal potential, $V_{rev}$, of the optocapacitive current ($N = 3$ biological replicates from fresh preparations in the presence of Gd³⁺). **d** Optocapacitive currents elicited by UV and blue light from a cell held at −60 mV and in the presence of 50 μM Gd³⁺. The time course of capacitance change was calculated according to Eq. 8 with $V_{rev} = V_s$.

effects[8]: (i) heating by blue light would have induced an increase in $C$, not the demonstrated decrease, and (ii) prolonged exposure to UV and blue laser light did not lead to the prolonged changes in $C$ that characterize photothermal events[8]. Further, the very symmetry of the magnitude of the changes in $C$ with UV and blue light exposure in Fig. 5d precludes a thermal effect. This is because heating with UV and/or blue light would always positively add to the change in $C$ that is due to photolipid switching, thus increasing the increment in $C$ with UV and reducing the decrement in $C$ with blue light exposure; this we do not observe.

From the above experiments, we infer a molecular mechanism for depolarization following capacitance-decreasing and initially hyperpolarizing photolipid photoisomerization. Specifically, blue light gives rise to mechanical membrane tension. Mechanosensitive ion channels open in response to this stimulus, producing a depolarization that evokes an AP.

It is important to note that other factors may have contributed to the blue light-driven augmented cellular excitability in addition to tension. Since ion channels are susceptible to the physical state of the embedding matrix[25], the photo-induced changes in membrane material properties such as bilayer surface area and thickness[17–19], bending rigidity and area compressibility[19,52,53], as well as the propensity towards domain formation[20,22] may also play a role. For example, changes in bilayer thickness and stiffness regulate the prominent bacterial mechanosensitive channels MscL and MscS[23,54], and membrane mechanical calculations predict that bilayer bending rigidity contributes to Piezo1 gating[55].

Yet, whilst bilayer material properties are bound to affect mechanosensitive channels[25], the primary stimulus for their gating is stretch activation by membrane tension[24]. Membrane tension—'force from lipids'—opens the two-pore domain potassium ion channels TRAAK and TREK1[56], the bacterial osmotic emergency valve MscL[39], as

well as the eukaryotic stretch-activated channel Piezo1[57]. The effect of lipid-transmitted tension must be distinguished from specific lipid-mediated effects—as has been established for TREK1's gating that may be enhanced by binding of anionic lipids[58].

## A photolipid-based approach to phototrigger direct depolarization and AP generation in excitable cells

We propose a photolipid-based approach to induce direct depolarization and AP generation in excitable cells by light. The approach does not require the presence of intermediary mechanosensitive channels that would translate a light-induced increase in membrane tension into depolarizing currents. Additionally, upon blue light exposure, Opto-DArG causes initial hyperpolarization of the membrane potential. We find that in the presence of mechanosensitive channels, the lateral stress generated upon transitioning from cis- to trans-OptoDArG may trigger depolarizing mechanosensitive currents, causing delayed depolarization. The recently developed photoswitch Ziapin2[29] induces the same response upon blue light excitation. Yet, only OptoDArG is fundamentally applicable to exert bidirectional direct optocapacitive control over the membrane potential.

Whilst light power and photolipid concentration allow for adjustable modulation of the membrane potential, it is worth noticing that we used high irradiances of up to 4 kWcm$^{-2}$ (UV) and 15 kWcm$^{-2}$ (blue) to evoke APs. Yet, trigger pulses on the order of only 1 ms were sufficient to elicit APs (Fig. 4). Consequently, with only 4–15 Jcm$^{-2}$, the doses—irradiance times exposure duration—used were relatively low, which should mitigate unwanted side effects. For comparison, calcium waves in HEK cells were stimulated by exposing molecular motors to 320 Wcm$^{-2}$ of 400 nm light for typically 250 ms[59]. That is, the underlying activation of inositol triphosphate signaling pathways used a total dose of 80 Jcm$^{-2}$. Further, triggering APs in neurons in the presence of gold nanoparticles by a photothermal optocapacitive approach required 31 kWcm$^{-2}$ × 1 ms = 31 Jcm$^{-2}$ of 532 nm light[9]. That is, the use of photolipids appears advantageous as the photoeffects are induced at a fraction of the energy injected by comparable methods. By delivering light in an even shorter interval in the μs-time range[10], the dose to elicit an AP by the optocapacitive approach can be reduced even further. Beyond the proof of concept demonstrated here, this emphasizes that the photolipid-based approach can be improved in the future, e.g., by more efficient photolipids.

The photolipid-based optocapacitive approach may be readily implemented considering that externally added OptoDArG spontaneously inserts into both leaflets of excitable cells, providing an elegant alternative method to genetic manipulations. Yet, further studies are required to understand the complexity of physiological effects evoked by photolipid photoisomerization in complex biological membranes. Effectively, our observation that photolipid photoisomerization can trigger mechanosensitive channels constitutes one "side effect" of our photolipid-based optocapacitive approach.

## Methods
### Horizontal planar lipid bilayer experiments
*E. coli* Polar Lipid Extract (PLE) was obtained from Avanti Polar Lipids (distributed by Merck). OptoDArG was synthesized as described previously[16]. Lipid aliquots and mixtures were prepared within amber glass micro reaction vessels from lipids dissolved in chloroform. Prior to storage at −80 °C, the solvent was evaporated by a mild vacuum gradient (Rotavapor, Büchi Labortechnik AG), and the dried lipids were flooded with argon.

Horizontal solvent-depleted planar lipid bilayers (PLBs) were folded from lipid monolayers on top of aqueous buffer in the lower and upper compartment of a custom-built chamber assembly made from PTFE[20,60]; the buffer used for all PLB experiments was 150 mM KCl, 10 mM HEPES, pH 7.4. First, an aperture between 70 to 100 μm in diameter in 25 μm-thick PTFE foil (Goodfellow GmbH) was created by

high-voltage discharge. Following pretreatment of the septum with 0.6 vol% hexadecane in hexane, hexane was allowed to evaporate for >1 h, except for the experiment shown in Fig. 2b where the septum was applied shortly after pretreatment to enforce a large torus. The torus is a solvent-containing annulus that eventually anchors the PLB within the circular aperture in the PTFE septum (schematically depicted in Fig. 1b) and effectively accommodates the several nm-thick PLB within the 25 μm-thick septum[61]. The presence of a torus, also called Plateau-Gibbs border, is necessary for the formation of stable PLBs[62] and we have observed that its extent may vary depending on the amount of residual hexadecane during folding. Then, the septum was attached by silicon paste to the lower side of the upper compartment of the chamber assembly. Lipid monolayers were prepared by applying lipid mixtures dissolved in hexane at a concentration of 10 mg/mL onto the aqueous interfaces of the upper and lower compartment. Unless noted otherwise, the lipid mixture was 80 m% *E. coli* PLE + 20 m% OptoDArG. After the hexane had evaporated, the folding of PLBs in a horizontal configuration was achieved by rotation of the upper compartment of the chamber assembly.

A 30 mm-diameter cover glass (No. 1, Assistent, Hecht Glaswarenfabrik GmbH & Co KG) fixed with a threaded PTFE ring comprised the bottom of the lower chamber. The chamber holder was installed on the sample stage of an Olympus IX83 inverted microscope equipped with an iXon 897 E EMCCD (Andor, Oxford Instruments Group). To position the horizontal PLB within working distance of a 40 × magnification (1.30 NA) infinity-corrected plan fluorite oil immersion objective (UPLFLN40XO/1.30, Olympus), the chamber holder was equipped with screws for achieving fine translation of the upper compartment in z-direction. The real-time controller (U-RTC, Olympus) used for synchronizing lasers and electrophysiological acquisition, as well as the motorized microscope, were controlled using the proprietary cellSens software (Olympus).

For electrical measurements, Ag/AgCl agar salt bridges containing 0.5 M KCl were put into the compartments and connected to the headstage of an EPC 9 patch-clamp amplifier (HEKA Elektronik, Harvard Bioscience). The headstage and chamber assembly were housed in a Faraday cage. Voltage-clamp and software lock-in measurements were controlled using PATCHMASTER 2x91 software (HEKA Elektronik, Harvard Biosciences). In both recording modes, current was analogously filtered at 10 kHz by a combination of Bessel filters and acquired at 50 kHz. Sine wave parameters for software lock-in measurements ("Sine+DC" method with computed calibration) were 10–20 mV peak amplitude, 5 kHz, 10 points per cycle, no averaging; voltage offset was ±10 mV.

For cis to trans photoisomerization of OptoDArG-containing PLBs requiring blue light, the output beam of a 488 nm diode laser (iBEAM-SMART-488-S-HP, TOPTICA Photonics) was cleaned-up by a ZET488/10x excitation filter (Chroma), directed through a Keplerian-type beam expander with pinhole, and eventually focused into the back-focal plane of the objective via the IX83's ZET488/640rpc main dichroic (Chroma). The diameter of the laser profile at the sample stage was ≈58 μm (1/e²). At a software-set output power of 200 mW, 30 mW exited the microscope objective, as determined by a photodiode (S120VC, Thorlabs).

For trans-to-cis photoisomerization requiring UV light, the output beam of a 375 nm diode laser (iBEAM-SMART-375-S, TOPTICA Photonics) was expanded using a separate Keplerian-type beam expander, focused into the back-focal plane of the objective via a ZT458RDC (Chroma) and the IX83's ZET488/640rpc main dichroic. The diameter of the laser profile at the sample stage was roughly 150–250 μm; ≈30 mW exited the microscope objective at a software-set output power of 70 mW.

Electrophysiological data recorded with PATCHMASTER were exported and analyzed using Mathematica (Wolfram Research); the presented graphs were created in OriginPro 2023b (OriginLab

Corporation). Integration of optocapacitive currents obtained at different clamped voltages and subsequent linear fitting was done by default to infer capacitance change and voltage offset (e.g., Fig. 2g and 2h); recordings indicating a voltage offset $> \pm 20$ mV for symmetric bilayers were excluded from further analysis.

## Patch-clamp experiments

Whole-cell patch clamp experiments were performed according to Hamill, et al. [63]. We used an in-house designed and built patch clamp amplifier for both current and voltage-clamp experiments. Data was sampled at 1 MHz, digitally filtered at Nyquist frequency, and decimated for the desired acquisition rate. To acquire the data, we used in-house software (Gpatch64MC) to control the 16-bit A/D converter (USB-1604, Measurement Computing, Norton, MA). The current and voltage signals were sampled at 100 kHz and filtered at 5 kHz. The current signal was filtered using a 4-pole Bessel filter (FL4, Dagan Corporation, Minneapolis, MN) and the voltage signal was also filtered using a 4-pole Bessel filter (Model 900, Frequency Devices, Haverhill, MA). Borosilicate patch pipettes (1BF120F-4, World Precision Instruments, Sarasota, FL) were pulled on P-2000 (Sutter Instruments, Novato, CA) horizontal puller, and the resistance ranged from 2–5 MΩ when back-filled with pipette solution. The pipette and bath solutions were based on Carvalho-de-Souza, et al.[9] and their composition were (mM): The internal (pipette) solution NaCl 10, KF 130, MgCl$_2$ 4.5, HEPES 10, EGTA 9, and pH 7.3 (KOH) and the external solution NaCl 132, KCl 4, MgCl$_2$ 1.2, CaCl$_2$ 1.8, HEPES 10, glucose 5.5 and pH 7.4 (NaOH). In some cases, to increase the driving force for Na$^+$ ions, we replaced NaCl from the pipette solution to KCl, and increased NaCl on the external to 140 mM. Recordings were performed at room temperature ($\approx$17–18 °C). When necessary, we used solutions containing GdCl$_3$ (439770, Sigma-Aldrich, St. Louis, MO) from 30 mM stock solutions dissolved in water. This stock solution is then diluted to the bath solution at the desired final concentration. Analysis was performed using in-house software. Analysis and graphs were constructed using OriginPro 2023b (Origin Lab Corporation, Northampton, MA, USA).

## Optical stimulation setup

We used a Ti:Sapphire (Mai Tai HP, Spectra-Physics, Milpitas, CA) laser combined with a multi-harmonic generator (ATsG-3-0.8-P, Del Mar Photonics, San Diego, CA) to obtain 367 nm laser wavelength for UV light illumination. Non-converted IR light exiting the harmonic generator was rejected by using 350–400 nm dielectric mirrors (BB1-E01, Thorlabs) and an LP400 longpass filter to reflect the UV light to the sample. For blue light illumination we used a laser diode with 447 nm (ams OSRAM) mounted on a cage (CP1LM9, Thorlabs, Newton, NJ) with an aspheric lens (LDH9-P2, Thorlabs). To allow maximum UV light transmission, we deployed a fused silica objective designed in-house to focus the laser beams on the cell and to image the cells. The diameter of the UV laser at the sample stage was $\approx$62 μm and of the blue laser was $\approx$ 36 μm. Since the deployed HEK cells had a typical diameter between 18 to 25 μm, the lasers illuminated the whole cell. The UV and blue laser beams were aligned to be at the same spot. The irradiance is estimated to be 4 kW/cm$^2$ for the UV laser and 15 kW/cm$^2$ for the blue laser, calculated considering the diameters of the beams and the full power used in this study, 124 mW and 160 mW, respectively. We used a camera (DCC1545M-GL, Thorlabs) mounted at an in-house built microscope to visualize the cells. We utilized a shutter (VS25S1ZO, Vincent Associates, Rochester, NY) to get onset laser pulses in <15–20 μs by using a Keplerian telescope to focus the beam before the shutter. We used custom-designed electronic circuitry to achieve fast onset of laser pulses with the blue laser diode. Using a camera, we placed the cells at the center of the laser beams using the blue laser in low power mode as a guide.

## Cell labeling

To label the HEK293 cells with the OptoDArG we followed the protocol established by Leinders-Zufall, et al. [64]. Briefly, prior to labeling, the stock solution containing 60 mM of OptoDArG, dissolved in dimethyl sulfoxide (DMSO, 276855, Sigma-Aldrich), was heated for 5 min at 37 °C. The stock solution was diluted with recording/bath solution to the final concentration of 60 μM. In some cases, to increase the amount and efficiency of labeling, we used 120 μM OptoDArG (indicated in the text when used). The solution was then vortexed and incubated at 37 °C for 5 min. The dish containing the cells was taken from the incubator and washed 1–3 times with the external solution. After washing the dish containing the cells, we added the solution containing OptoDArG and incubated it at 37 °C for 1 h. After labeling, we washed the dish 1–3 times with the external solution and assembled it in the recording chamber. The stock solution containing the OptoDArG was aliquoted in small volumes ($\approx$6 μl) and stored at −20 °C. The DMSO concentration never exceeds 0.2% in the experiments.

We considered the OptoDArG labeling (incorporation) successful when we observed outward currents at −70 mV elicited by the blue laser. This rules out thermal optocapacitive effects induced by blue light exposure since heating would cause d$C$/d$t > 0$ and this would result in an inward (negative) current (Eq. 3). Indeed, the generation of outward currents at negative holding potentials was a hallmark of the photolipid (OptoDArG)-based optocapacitive approach (Supplementary Fig. 4c). When the magnitude of the (negative) inward current induced by the UV laser was >200 pA, we considered a success in the labeling. When a small amount of photolipids was incorporated (inward current magnitude of <200 pA), we did not try to elicit an AP because not enough depolarization for eliciting an AP would have been generated. Confirming the results obtained without photolipid, it can be observed in Supplementary Fig. 4 that when the incorporation procedure failed, no optocapacitive current was generated at −70 mV by UV–both in the presence and absence of Gd$^{3+}$ in the bath. Therefore, we did not proceed with current-clamp experiments, as detailed in Supplementary Tables 1 and 2. The success rate of labeling with OptoDArG was approximately the same in the presence (56.6%) or absence of Gd$^{3+}$ (59%). These values are similar to those reported in literature[64]. This similarity is expected since Gd$^{3+}$ was not present during the incorporation procedure: it was added after the labeling with OptoDArG was finished.

## Cell culture

HEK293 cells stably transfected with Na$_V$1.3 channels were purchased from American Type Culture Collection (CRL-3269 (currently discontinued), ATCC, Manassas, VA). The cells, at the time they were acquired, were expressing the Kir2.1 K$^+$ channel. However, after 7–8 passages, the cells lost the ability to express Kir2.1. Therefore, all experiments were performed under the condition that no Kir2.1 was present, as it can be observed in the representative Na$^+$ current traces shown in Supplementary Fig. 4e. They were cultured in 75 cm$^2$ flasks in a humidified incubator at 37 °C with 5% CO$_2$, following ATCC protocols. The cells were grown and maintained in DMEM/F12 medium (ATCC 30-2006) supplemented with 10% heat-inactivated fetal bovine serum (FBS, SH30071.03, HyClone-GE healthcare), 0.5 mg/ml G-418-sulfate (10131-027, Gibco, ThermoFisher), 100 U/ml penicillin-G sodium and 100 μg/ml streptomycin sulfate (P4333, Sigma-Aldrich) and 2.0 μg/mL Puromycin (A11138-03, Gibco). Dr. Eduardo Perozo kindly provided HEK293T ATCC (CRL-1573) cells, and they were grown in Dulbecco's modified Eagle medium (DMEM, 11995065, Gibco) supplemented with 10% FBS and 100 U/ml penicillin-G sodium and 100 μg/ml streptomycin sulfate. They were cultivated at 37 °C in 5% CO$_2$ humidified incubator. Before experiments, cells were detached using Accutase (AT-104, Innovative Cell Technologies, Inc., San Diego, CA) following the manufacturer's protocol. They were centrifuged for

5 min at 125 × g; the supernatant was replaced by fresh medium and seeded (≈30–50% confluency) into previously prepared Poly-L-lysine treated culture dishes. They were used within 12–36 h for electrophysiological experiments. Glass-bottomed culture dishes (D35-10-1-N, Cellvis, Mountain View, CA) were incubated with Poly-L-lysine solution (P8920, Sigma-Aldrich) for 15 min at room temperature and thoroughly rinsed with sterile PBS (10010-023, Gibco) and stored until use.

## Reporting summary

Further information on research design is available in the Nature Portfolio Reporting Summary linked to this article.

## Data availability

The data that support this study are available from the corresponding authors upon request. A Source Data file pertaining to Figs. 1, 2, 3, 5 and Supplementary Figs. 1–3 is provided.

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

## Acknowledgements

We want to thank Drs. Navid Bavi and Eduardo Perozo for their insightful discussions and suggestions. This work is supported by the Austrian Science Fund (FWF) Award P34826 to P.P., National Institutes of Health Award R01GM030376 to F.B., and National Science Foundation Award QuBBE QLCI (NSF OMA-2121044) to F.B.

## Author contributions

C.B. conducted and with F.B. analyzed the cell experiments. J.P. and R.Y. conducted and with S.S. and P.P. analyzed the planar lipid bilayer experiments. T.G. synthesized OptoDArG. All authors contributed to designing the research and writing the paper.

## Competing interests

The authors declare no competing interests.
