## [Peer Review File · Nature Communications]

Photolipid excitation triggers depolarizing optocapacitive currents and action potentialsReviewer #1 (Remarks to the Author):

In their manuscript "Photolipid excitation triggers depolarizing optocapacitive currents and action potentials" Bassetto and co-authors report on the usage of an azobenzene based lipid (OptoDARg) to trigger changes in local membrane capacities that lead to current flows in their measurement device, and later to action potentials in mammalian cell culture. The paper is very well written and structured in two main parts. In the beginning, the authors demonstrate that illuminating reconstituted membranes from lipid extracts that are mixed with a high mass fraction (20m%) of OptoDARg results in very impressive and well modeled changes in membrane capacitance. Here the hypothesis is drawn that also the membrane tension is changed, which is a result that turns out to be important later when applying the molecule to living cells. In the second part, the authors combine the OptoDARg with HEK293 cells, that have also been transfected with a sodium channel to mimic the possibility of showing actin potentials. The effect of the OptoDARg is then claimed to be twofold. One effect is via the change in membrane potential and a secondary effect is via an increase in membrane tension. Here extremely strong laser fields were used, which raise serious doubts about the effect of the photolipids in comparison to the known effect such lasers have. Additionally, this part suffers from statistics.

Overall, this is an interesting study that investigates the potential of a well-known and commercially available compound. While the section studying the effects of the photolipid switching on a reconstituted membrane is convincing, I have serious concerns about the experimental approach toward the cells. Additionally, many experimental details are missing, which makes it very difficult to fully understand the experimental procedures. Therefore I cannot recommend publication of the paper in the current form.

Main concerns:

- 1) The authors need to provide more detail on the experiments. For example, I could not connect the sketch of the experiment in Figure 1b with the mentioned torus in figure 2a and b. What do we see here? How is a torus embedded in the design of figure 1b? Where would the mentioned lipid reservoir be? In Figure 2 it seems that the torus (hole??) and the membrane are disconnected. This needs more details to be understood by a interested reader.
- 2) Additionally, the authors fail to describe the optical stimulation of the cells sufficiently. Are the pulsed and UV lasers homogenously illuminating the full field of view? Are they focused by a microscope objective? How large is the area illuminated by the lasers? How are cells targeted by the laser? What is the intensity at the cell (for this we need the illuminated area). If a large area illumination was used, how do the authors deal with the long coherence length of the lasers and hence the resulting speckles (in both setups)?
- 3) I cannot fully follow the explanation about the relaxation of C in the size-dependent experiments (Fig 2ab). If the illuminated area is always the same, I can understand that in the larger case the relaxation is quicker as there is more connection to the reservoir. But why would in this case the overshoot be larger? If there is a larger membrane area compared to the illuminated area in the larger torus, I would assume that the overshoot becomes smaller as the effect of tension change is already partially buffered by the non-illuminated lipid area, which could relax the tension immediately. More details on the reasoning here are required.
- 4) Why are two different setups used? I suggest doing the cell experiments with the same light source and experimental setup as the membrane experiments. The different setups question if the results from the lipid extracts can be transferred to the cells.
- 5) I have a serious problem with the illumination of the cells. First, using a pulsed laser leads to extremely high EM-Fields. Did the authors filter out the non-converted component (the harmonic generator will only convert part of the IR light)? Even if they did, the UV pulse would also be very short. It is known in the literature (and mentioned by the authors) that similar light intensities can easily lead to the excitation of action potential or local ruptures in the membrane. Having said this, the cell experiments suffer from statistics and convincing controls as these lasers provide ample possibility for wrong interpretations of the resulting data:
- 5A) Regarding statistics, we need first to know how the cells were illuminated, and how was ensured

that always the same light intensity was deposited on the cell (how did the experimenter 'aim' at the cell). Where on the cell was the light deposited? Always at the same region, over the full cell, at the membrane edge?

5B) As shown in Fig. 4e not all illumination led to a depolarization, hence we need to get more data regarding the statistics of AP generation. How often was depolarization successful, and how does this compare to the situation without OptoDARG? This needs to be done for the different cases of cell with and without the photolipid, and with/without Gd. I would like to point out that these experiments are critical to be convincing with respect to the claims. I am aware that the experiments on the cells are challenging. Still, to be really convincing these statistics are vital.

5C) More details about the experimental and biological replicates are needed. Especially in Figure 4 and 5. Best is to provide a statistical quantification for each claim and then give the numbers of replicates in the figure caption.

5D) The authors need to apply a statistical test for the capacitance decay that they determine in figure 3f. Given the spread of the data in figure 3f, I am unsure if the reported trend is statistically significant. Also, we need to know if the 10 cells have been recorded in a single session, or if these are biological replicates from fresh preparations.

5E) We need more quantification of the control, i.e. data without the photolipid. At the moment there is a single trace shown. Given many reports that at similar pulsed laser systems, the thermal effects can lead to currents, this is a sensitive issue. Please demonstrate that the AP is not simply triggered by the lasers, but by the combination of lasers and photolipid. Again, this is important as it is a key claim of the paper.

6) The explanation for the action potential using blue light as an effect of membrane tension is very interesting. The authors can here include recent work that shows an increase in the area using another azobenzene molecule (Georgiev et al, 2018, Advanced Science), although another paper has recently shown an the opposite effect on membrane area (Hoglsperger 2023, Nat. Comm). The simple calculation presented suggests that enough energy is provided to switch mechanosensitive channels. However, this calculation is done for a large membrane area. Following the logic, the distance between triggering channels would be in the order of $6\mu\text{m}$ (the calculation was done for $(40\mu\text{m})^2$ area)...can this be realistically leading to a trigger of tension-mediated AP? Additionally, the expected m% of the photolipid in the cells may be much smaller than in the PLB experiments used for the estimation.

Minor comments:

- 1) Cell viability: The authors should quantify possible cell death due to the application of the photolipid.
- 2) The HEK cells are not listed anymore at ATCC. It would be useful to mention this.
- 3) The stably transfected HEK cells also express an additional potassium channel. To which extend can this additional channel interfere with the results reported.

Reviewer #2 (Remarks to the Author):

The manuscript introduces a method for regulating neuronal activity using photolipids, specifically azobenzene-containing diacylglycerols, without the need for genetic modifications. These photolipids mediate light-triggered cellular de- or hyperpolarization. Upon exposure to UV light, photolipid conformation alters, embedding plasma membranes with increased capacitance and allegedly inducing depolarizing currents, which subsequently trigger voltage-gated sodium channels in cells, leading to action potentials. In contrast, blue light decreases membrane capacitance, allegedly causing hyperpolarization. This approach allows for precise cellular excitation through controlled illumination and self-insertion of photolipids into membranes.

While the manuscript offers an insightful exploration into the capabilities of OptoDARG as a light-switch molecule, it would be significantly beneficial to readers if the authors could provide a comparative analysis with previously reported molecules possessing similar mechanisms of action. In particular, the

molecule Ziapin2, as detailed in *Nature Nanotechnology* 15, 296–306 (2020), and the light-activated molecular machines described in *Nature Nanotechnology* 18, 1051–1059 (2023), stand out as pertinent comparatives. Drawing parallels and contrasts with these molecules would not only underscore the novelty or advantages of OptoDARg but also offer a comprehensive landscape of the advancements with pros/cons analysis in the field. I would strongly urge the authors to incorporate this comparison to enrich the discourse and contextual relevance of their findings.

I would like some clarification regarding the sign of the transmembrane optocapacitive current, I_{cap} , presented in the paper. Specifically, referring to equation (3), it suggests a negative, hyperpolarizing current when V is negative, such as $V = -130$ mV during UV light illumination, especially considering dC/dt is positive during membrane thinning under UV illumination. A negative current, which effectively increases the charge on the capacitor with an increasing capacitance due to membrane thinning, should produce a hyperpolarizing effect. Yet, in Figures 3 and 4, the UV light appears to have a depolarization effect, suggesting a positive I_{cap} under UV illumination. Please clarify.

A significant point of concern arises from the absence of characterizing local temperature changes during illumination. While the authors assert that the photoswitching molecule presents a distinct mechanism from the traditional photothermal-based optocapacitive mechanisms, the lack of temperature measurements during illumination undermines this argument. Without these crucial measurements, the possibility remains that there may still be photothermal effects (especially at high intensities such as shown in Fig. 1e) contributing to the observed changes, even if unintentional.

In examining Figures 1c and 1d, I observed distinct disparities in the timescales of the optocapacitive currents between the cis-to-trans and trans-to-cis transformations. Specifically, the trans-to-cis transformation depicted in Figure 1d seems to manifest at a considerably slower pace (almost 10 ms) compared to the cis-to-trans transformation in Figure 1c (1–2 ms). Could the authors elucidate the underlying reasons for this pronounced difference? This point is worth discussing especially because the authors claim that the millisecond level current under UV light (which is actually close to 10 ms) has a depolarizing effect to trigger action potentials.

Reviewer #3 (Remarks to the Author):

The work reports on a new method for modulating the electrical behavior in excitable cells by using light and photolipids. The work is clear, well written and the topic is of high interest. The reported measurements are of high quality and the analysis of the results look comprehensive. In light of the above I recommend publication after minor revisions.

- The authors suppose that permittivity of the membrane is constant upon light exposure, mechanical stress and rearrangement (see equation 1 and the following analysis through the whole text). Hence, they base the whole data analyses on the variation of membrane capacitance due to membrane thickness and area. Indeed, this is not proved. Molecular polarizability may change upon rearrangement of lipid layers and, for example, water or salt ions solvation may change and/or ions may penetrate in between lipid molecules. This may significantly alter the polarizability and then the permittivity. I agree that this is expected to be a minor contribution however, as far as I know, this has never been investigated and then proven. Hence, I recommend to the authors to clearly state that their analyses are based on this assumption unless they can show the opposite or cite previous literature.

- It's worth noticing that the light power used to switch the isomer conformation is in the order of 100 Watt per cm^2 in the UV region (in the visible is even higher). This is a very high value that may compromise practical applications. I think that the approach can be improved and optimized in the future and the aim of the present work is to prove the concept and its functioning. However, the author should better highlight this crucial point in the text and, in particular in the conclusion.

- Still regarding the discussion and the conclusion I would pay attention to state that the approach "it is easy to handle since externally added photolipids spontaneously insert into both leaflets of excitable

cells". I agree, however, the lipid membrane is the result of 1 billion year of evolution. It does play a fundamental and very complex role in cell biology. Its modification with no side effects it is not trivial to achieve or to predict. Hence, I would be more cautious, and I would clearly say further studies are necessary to ensure that there are no side effects due to membrane modifications.

Below, the reviewers' comments are reproduced with our replies interspersed in **[[brackets in blue type]]**. In **highlight** is the text added to the manuscript. Quotations from the manuscript are in "quotation marks".

Reviewer #1

In their manuscript "Photolipid excitation triggers depolarizing optocapacitive currents and action potentials" Bassetto and co-authors report on the usage of an azobenzene based lipid (OptoDARg) to trigger changes in local membrane capacities that lead to current flows in their measurement device, and later to action potentials in mammalian cell culture. The paper is very well written and structured in two main parts. In the beginning, the authors demonstrate that illuminating reconstituted membranes from lipid extracts that are mixed with a high mass fraction (20m%) of OptoDARg results in very impressive and well modeled changes in membrane capacitance. Here the hypothesis is drawn that also the membrane tension is changed, which is a result that turns out to be important later when applying the molecule to living cells. In the second part, the authors combine the OptoDARg with HEK293 cells, that have also been transfected with a sodium channel to mimic the possibility of showing actin potentials. The effect of the OptoDARg is then claimed to be twofold. One effect is via the change in membrane potential and a secondary effect is via an increase in membrane tension. Here extremely strong laser fields were used, which raise serious doubts about the effect of the photolipids in comparison to the known effect such lasers have. Additionally, this part suffers from statistics.

Overall, this is an interesting study that investigates the potential of a well-known and commercially available compound. While the section studying the effects of the photolipid switching on a reconstituted membrane is convincing, I have serious concerns about the experimental approach toward the cells. Additionally, many experimental details are missing, which makes it very difficult to fully understand the experimental procedures. Therefore I cannot recommend publication of the paper in the current form.

[[We appreciate the extensive feedback and the questions raised by the Reviewer. Please see below our responses in blue.]]

Main concerns:

1) The authors need to provide more detail on the experiments. For example, I could not connect the sketch of the experiment in Figure 1b with the mentioned torus in figure 2a and b. What do we see here? How is a torus embedded in the design of figure 1b? Where would the mentioned lipid reservoir be? In Figure 2 it seems that the torus (hole??) and the membrane are disconnected. This needs more details to be understood by a interested reader.

[[We have modified Fig. 1 to show the localization of the torus and we have added additional information on its role and position including relevant references that allow for obtaining deeper insights in the *Methods* section.

page 16: "Following pretreatment of the septum with 0.6 vol% hexadecane in hexane, hexane was allowed to evaporate for > 1 h, except for the experiment shown in Fig. 2b where the septum was applied shortly after pretreatment to enforce a large torus. **The torus is a solvent-containing annulus that eventually anchors the PLB within the circular aperture in the PTFE septum (schematically depicted in Fig. 1b) and effectively accommodates the several nm-thick PLB within the 25 μm-thick septum⁶¹. The presence of a torus, also called Plateau-Gibbs border, is necessary for the formation of stable PLBs⁶²**

and we have observed that its extent may vary depending on the amount of residual hexadecane during folding.”

In Fig. 2a/b, the planar lipid bilayer appears transparent due to its nanometer thickness whilst the visually separated torus surrounding it increases from several nanometers to micrometers, i.e. the thickness of the septum – both are within the aperture within the PTFE septum. Surrounding the torus is the bulk of the 25 μm -thick PTFE septum. We have labeled lipid bilayer, torus and septum in Fig. 2a/b to clarify these regions.]]

2) Additionally, the authors fail to describe the optical stimulation of the cells sufficiently. Are the pulsed and UV lasers homogeneously illuminating the full field of view? Are they focused by a microscope objective? How large is the area illuminated by the lasers? How are cells targeted by the laser? What is the intensity at the cell (for this we need the illuminated area).

[[We now provide more detailed information on the optical setups used for stimulating cells and updated Fig. 3a to better represent the experimental conditions of our setup. Before experiments, we aligned the two laser beams (UV and blue) to be at the same spot and measured the power. Both lasers are not illuminating the full field of view, and we are not using a beam homogenizer. The diameter of the UV laser is $\approx 62 \mu\text{m}$ and the blue is $\approx 36 \mu\text{m}$, and usually cells have a diameter between 18 to 25 μm . Thus, the lasers illuminated the whole cell, and they are focused on the cell. The irradiance is estimated to be 4 kW/cm^2 for the UV laser and 15 kW/cm^2 for the blue laser, calculated considering the diameters of the beams and the full power used in this study, i.e. 124 mW (UV) and 160 mW (blue), respectively. The laser beam is targeted on the cells by a fused silica objective designed and built in-house, as stated in the *Methods* section. This same objective is used to observe the cells. We modified the *Methods* section to clarify experimental details missing for the experiments carried out on cells.

page 17: “Optical stimulation setup

We used a Ti:Sapphire (Mai Tai HP, Spectra-Physics, Milpitas, CA) laser combined with a multi-harmonic generator (ATsG-3-0.8-P, Del Mar Photonics, San Diego, CA) to obtain 367 nm laser wavelength for UV light illumination. Non-converted IR light exiting the harmonic generator was rejected by using 350–400 nm dielectric mirrors (BB1-E01, Thorlabs) and an LP400 longpass filter to reflect the UV light to the sample. For blue light illumination we used a laser diode with 447 nm (ams OSRAM) mounted on a cage (CP1LM9, Thorlabs, Newton, NJ) with an aspheric lens (LDH9-P2, Thorlabs). To allow maximum UV light transmission, we deployed a fused silica objective designed in-house to focus the laser beams on the cell and to image the cells. The diameter of the UV laser at the sample stage was $\approx 62 \mu\text{m}$ and of the blue laser was $\approx 36 \mu\text{m}$. Since the deployed HEK cells had a typical diameter between 18 to 25 μm , the lasers illuminated the whole cell. The UV and blue laser beams were aligned to be at the same spot. The irradiance is estimated to be 4 kW/cm^2 for the UV laser and 15 kW/cm^2 for the blue laser, calculated considering the diameters of the beams and the full power used in this study, 124 mW and 160 mW, respectively. We used a camera (DCC1545M-GL, Thorlabs) mounted at an in-house built microscope to visualize the cells. We utilized a shutter (VS25S1ZO, Vincent Associates, Rochester, NY) to get onset laser pulses in less than 15–20 μs by using a Keplerian telescope to focus the beam before the shutter. We used custom-designed electronic circuitry to achieve fast onset of laser pulses with the blue laser diode. Using a camera, we placed the cells at the center of the laser beams using the blue laser in low power mode as a guide.”]]

If a large area illumination was used, how do the authors deal with the long coherence length of the lasers and hence the resulting speckles (in both setups)?

[[In both cases, the lasers have a profile of several tens of μm at the sample. Whilst our profiles show a few μm -large inhomogeneities, the size of our targets, cells and planar lipid bilayers, is on the order of tens of μm . Hence, we ensured that the lasers illuminate the samples in their entirety. These local inhomogeneities are of little consequence to the optocapacitive mechanism as indicated by the observation that the generated currents are sufficiently well described by a monoexponential fit (cf. Eq. 7). The result of the monoexponential fits to the optocapacitive currents shown in Fig. 1e allowed us to calculate irradiance-normalized rates of azobenzene photoisomerization that agree reasonably with literature values (Arya *et al.* *J Chem Phys.* **152**, 24904 (2020)). It also agrees with the concept of a “continuous” membrane tension over the lipid bilayer membrane. This concept is essential to the micropipette aspiration technique where pressure-driven aspiration of the lipid bilayer membrane of a giant unilamellar vesicle into a micropipette creates a protrusion and perturbs the tension of all contiguous vesicular membrane (see work on the microaspiration technique by Evan Evans, e.g., *Phys Rev Lett.* **64**, 2094–2097 (1990)). Hence, the Gaussian shape of the illumination profile and small local inhomogeneities were of little consequence under the deployed conditions.]]

3) I cannot fully follow the explanation about the relaxation of C in the size-dependent experiments (Fig 2ab). If the illuminated area is always the same, I can understand that in the larger case the relaxation is quicker as there is more connection to the reservoir.

[[The area of the two lipid bilayers shown in Fig. 2a/b differs by only $\approx 5\%$ as can be judged from the corresponding membrane capacitances. Yet, the size of the torus is vastly different, i.e., the lipid reservoir in Fig. 2b is much larger than in Fig. 2a. We did not attempt an investigation of the relaxation rates but looked at the relaxation amplitudes.]]

But why would in this case the overshoot be larger? If there is a larger membrane area compared to the illuminated area in the larger torus, I would assume that the overshoot becomes smaller as the effect of tension change is already partially buffered by the non-illuminated lipid area, which could relax the tension immediately. More details on the reasoning here are required.

[[The whole bilayer is illuminated, i.e., the $1/e^2$ -diameter of the blue laser covers nearly the whole PLB. Our interpretation of the results (Fig. 2a/b) is the following: Blue light exposure quantitatively decreases PLB area within a few milliseconds, leading to the build-up of membrane tension. C decreases to its minimal value. Then, lipids are drawn out of the torus, causing the relaxation of membrane tension and increase in C . In the presence of a large solvent-containing torus, more material can be pulled out of the torus and hence, the amplitude of fast relaxation is larger.

To that effect, we clarified our reasoning in the manuscript by rewriting the paragraph. We now explicitly denote ΔC as the change in capacitance upon photoisomerization within the first 3 ms (blue) or 10 ms (UV) after light exposure (see Eq. 8). That is, ΔC corresponds to the change in capacitance before the onset of fast relaxation (see, e.g., Fig. 2c/d). For clarity, we refer to the change in capacitance upon blue light-driven photoisomerization after fast relaxation as $\Delta\Delta C$ and indicate both ΔC and $\Delta\Delta C$ in Fig. 2a/b.

page 6f: “Upon closer inspection of the capacitance traces in Figs. 2a and 2b, we find that C drops to a minimum value within milliseconds (Fig. 2c) following blue light exposure (at 1.15 s) and then relaxes quickly; we refer to the change in C after relaxation relative to the initial level pre-exposure as $\Delta\Delta C$ (indicated in Figs. 2a and 2b). Consistent with the 3 ms integration time used to determine ΔC from I_{cap} (Eq. 8), ΔC corresponds to the drop in C within the first 3 ms following light exposure, as indicated in Figs. 2a and 2b. In the presence of a small torus, we observe that this fast relaxation in C (between 1.2 and 1.3 s in Fig. 2a, indicated by the gray arrow) amounts to $< 15\%$ of the amplitude of the drop, whilst it can reach $> 30\%$ in the presence of a larger torus (Fig. 2b; for details on how the large torus was

enforced, see Methods). Our interpretation of the results (Figs. 2a and 2b) is the following: Blue light exposure quantitatively decreases PLB area within a few ms, leading to a pulling force of the PLB on the torus and hence the build-up of membrane tension – C decreases to its minimal value (Fig. 2c). The force originates from area differences between cis and trans photolipids, as is explored further in the following section. Then, bilayer material is drawn out of the torus, causing at least a partial relaxation of membrane tension and the observed increase in C . In the presence of a larger solvent-containing torus, more material can be pulled out of the torus and hence, the amplitude of fast relaxation (i.e., $\Delta\Delta C - \Delta C$) is larger (Fig. 2b).

Interestingly, relaxation in C also follows the photolipid-mediated increase in C upon UV light exposure. Conceivably, it signifies that the constitutive tension of PLBs flattens photo-induced undulations or PLB bulgings that appear with the increment in membrane area upon membrane thinning. Importantly, these torus-specific relaxations do not represent an intrinsic feature of the photolipid. These minor drawbacks of the experimental model system notwithstanding, we find a general agreement between current and capacitance recordings. Applications of Eq. 3, Eq. 7 and Eq. 8 provide strong evidence for the optocapacitive mechanism of rapid photolipid photoisomerization.”]]

4) Why are two different setups used? I suggest doing the cell experiments with the same light source and experimental setup as the membrane experiments. The different setups question if the results from the lipid extracts can be transferred to the cells.

[[This paper emerged from a collaboration between two research groups with distinct but complementary expertise. The planar lipid bilayer experiments were conducted by Peter Pohl’s lab in Linz, Austria, and the cell experiments by Francisco Bezanilla’s lab in Chicago, USA. We believe that the fact that using our distinct experimental approaches and setups we nonetheless arrived at the same conclusions with regard to optocapacitive current generation and capacitance changes due to OptoDARg photoisomerization (compare Figs. 1 and 3) is a testimonial to the robustness and reproducibility of the presented method. For this reason, we do not think that the experiments should be repeated at the same setup.]]

5) I have a serious problem with the illumination of the cells. First, using a pulsed laser leads to extremely high EM-Fields. Did the authors filter out the non-converted component (the harmonic generator will only convert part of the IR light)?

[[We updated Fig. 3a. It now describes the setup used for cell stimulation in more detail. The UV light that leaves the harmonic generator is guided to the back of the objective by E01 mirrors from Thorlabs, that do **not** reflect IR (BB1-E01, https://www.thorlabs.com/newgrouppage9.cfm?objectgroup_id=139&pn=BB1-E01#2147). On top of that, there is a dichroic mirror that reflects the UV light and allows the blue light to pass (LP400). Therefore, there is no leak of IR light that reaches the cells and only UV light is present. We added a corresponding note to the *Methods* section:

page 17: “Non-converted IR light exiting the harmonic generator was rejected by using 350–400 nm dielectric mirrors (BB1-E01, Thorlabs) and an LP400 longpass filter to reflect the UV light to the sample.”]]

Even if they did, the UV pulse would also be very short. It is known in the literature (and mentioned by the authors) that similar light intensities can easily lead to the excitation of action potential or local ruptures in the membrane. Having said this, the cell experiments suffer from statistics and convincing controls as these lasers provide ample possibility for wrong interpretations of the resulting data:

[[Conceivably, the Reviewer refers to APs triggered by photothermal effects (*Neuron*. **86**, 207–217 (2015)). Here, we use an UV irradiance that is ≈ 10 -fold smaller and, thus, by itself does not promote heat-mediated effects. Note that the exposure times were very short (few ms; we were limited by the deployed mechanical shutter). Therefore, we can rule out the possibility of local heating that would excite action potentials or cause local ruptures in the membrane. The blue laser light can also be absorbed and if its power is larger than 200 mW, we start observing thermal effects, as demonstrated in the updated Supplementary Fig. 4c. The associated currents are larger than the optocapacitive current, i.e., overwriting it, and have the opposite sign because heating thins the membranes and thus $dC/dt > 0$, whereas cis-to-trans photoisomerization of OptoDARg with blue light causes $dC/dt < 0$ (Eq. 3). Therefore, in the experiments for this study the power of the blue laser **NEVER** exceeded 160 mW.

Please see the updated Supplementary Fig. 4 and its legend below that demonstrates the absence of effects and the lack of depolarization upon exposure of cells to the lasers in the absence of OptoDARg (no labeling (incorporation) with OptoDARg). The full description of the figure and more information about the cells can be found in the updated Supplementary information and Supplementary Tables 1 and 2 (see further below).

caption to Supplementary Fig. 4: “No optocapacitive current is present when cells were not labeled with OptoDARg and 5 μM Gd^{3+} does not affect Na^+ currents through Nav1.3 . a, Without OptoDARg labeling, no optocapacitive currents were induced by UV light exposure, indicating the absence of photothermal effects due to UV light itself. The red, green, and orange are representative current traces for cells that did not have OptoDARg, and the black for a cell that had OptoDARg for conditions in the absence (a) and in the presence of Gd^{3+} (b). In (a) the voltage was held at -70 mV and a single UV pulse was applied as indicated. In (b) the voltage protocol is inset at the top of the panel and the duration of exposure to laser light is shown by the rectangles. c, Thermal effects induced by blue laser when power exceeds 200 mW. The top trace (160 mW) shows light-induced capacitance changes by OptoDARg; the bottom trace (523 mW) displays the photothermal effect. Traces are from the same cell, and they were primed by UV prior to the indicated blue light exposure. d, No depolarization is observed when cells did not have OptoDARg incorporated. The red and black traces are recordings in voltage-clamp (top) and current-clamp (bottom) for cells in the absence and presence of OptoDARg, respectively. More details regarding the power used in our experiments can be found in Supplementary Table 1 and 2. The current and voltage-clamp experiments were from the same cell. e, Na^+ currents elicited by the voltage protocol inset in e, and in presence of 5 μM Gd^{3+} .”]

5A) Regarding statistics, we need first to know how the cells were illuminated, and how was ensured that always the same light intensity was deposited on the cell (how did the experimenter ‘aim’ at the cell). Where on the cell was the light deposited? Always at the same region, over the full cell, at the membrane edge?

[[The cells were illuminated and viewed using the same fused silica objective. We aligned both lasers and did not touch them after alignment. Prior to experiments, we measured the power and double-checked the alignment to ensure that it did not change. We used the blue laser in low power mode as a guide and used the camera to position the cell under investigation where the laser beam was set. Using the camera, we focused the lasers at the membrane which typically gives the maximum amplitude of the optocapacitive current. Because the size of the laser profiles at the sample stage is larger than the cells, the whole cell is illuminated. It is important to stress that we took the utmost care to carry out these experiments, such that we tried as much as we could to have the cells and the laser always at the same conditions. The deposited light intensity varied depending on the power used. The used power values are given in Supplementary Tables 1 and 2 (see further below).

page 17: “The diameter of the UV laser at the sample stage was ≈ 62 μm and of the blue laser was ≈ 36 μm . Since the deployed HEK cells had a typical diameter between 18 to 25 μm , the lasers illuminated the whole cell. The UV and blue laser beams were aligned to be at the same spot. The irradiance is estimated to be 4 kW/cm^2 for the UV laser and 15 kW/cm^2 for the blue laser, calculated considering the diameters of the beams and the full power used in this study, 124 mW and 160 mW, respectively. We used a camera (DCC1545M-GL, Thorlabs) mounted at an in-house built microscope to visualize the cells. To allow maximum UV light transmission, we deployed a fused silica lens designed in-house. We utilized a shutter (VS25S1ZO, Vincent Associates, Rochester, NY) to get onset laser pulses in less than 15–20 μs by using a Keplerian telescope to focus the beam before the shutter. We used custom-designed electronic circuitry to achieve fast onset of laser pulses with the blue laser diode. Using a camera, we placed the cells at the center of the laser beams using the blue laser in low power mode as a guide.”]

5B) As shown in Fig. 4e not all illumination led to a depolarization, hence we need to get more data regarding the statistics of AP generation. How often was depolarization successful, and how does this compare to the situation without OptoDARg? This needs to be done for the different cases of cell with and without the photolipid, and with/without Gd. I would like to point out that these experiments are

critical to be converging with respect to the claims. I am aware that the experiments on the cells are challenging. Still, to be really convincing these statistics are vital.

[[We understand the Reviewer’s concerns about the success in eliciting APs. To elicit an AP in a cell many factors play a role. For example, the amount of sodium channels expressed, the current carried by them and the level of inactivation that they have prior to firing an AP. Another factor is related to the input resistance of the cell. In our hands, these HEK293 cells expressing Nav1.3 required a large depolarization to elicit an AP and sometimes even with a depolarization of 30–35 mV cells failed to fire, as it can be seen in the figure below (new Supplementary Fig. 5). On the other hand, sometimes we were able to elicit an AP with depolarization of as little as 10 mV, likely due to different conditions of the cells, as described above. Therefore, only quantifying the success rate does not include all the information that influences the ability to generate an AP by the photolipids reported here. Nevertheless, we would like to point out that the experiments that the Reviewer asked for were already done prior to submission and a summary of them is now presented in the following two tables representing conditions in the absence (Supplementary Table 1) and presence of Gd³⁺ (Supplementary Table 2). Fig. 4 represents a summary of our findings. We incorporated the Tables 1 and 2 in the supplementary information so the readers will have as much information as possible about the 52 cells tested. We also added a text to discuss and clarify our results. Please note in the figure from question#5 (new Supplementary Fig. 4d) that there was no AP elicited by UV when OptoDARG was not incorporated in the cells.

page 12, caption to Fig. 4: “Whilst this figure serves as a summary of our findings, Supplementary Tables 1 and 2 contain experimental details on the 52 cells assayed (freshly prepared biological replicates).”

caption to Supplementary Fig. 5: “Cells expressing Nav1.3 required large depolarization to elicit an AP. The black trace represents an AP elicited by injecting current and the red trace failure to fire even though the depolarizations induced were similar. Both depolarizations were induced by injecting current.”

Supplementary Table1: Success of AP generation in the absence of Gd³⁺ and labeling conditions.

Recording file	peak current by UV at -70 mV (nA)	laser power		ΔV (mV) by UV	AP by UV	labeling	ΔV (mV) by injecting current
		UV (mW)	Blue				
Nav13_02-01--23-001_006_ch1.dat	-0.218201	100	152	10.68	No	Yes	not measured
Nav13_02-01--23-003_004_ch1.dat	-0.0427246	100	152	not tried	not tried	No	
Nav13_02-01--23-005_003_ch1.dat	-0.108337	100	152	7.02	Yes	Yes	not measured
Nav13_02-01--23-006_002_ch1.dat	-0.0869751	100	152	7.69	Yes	Yes	not measured

Nav13_02-01--23-007_007_ch1.dat	-0.0442505	100	152	not tried	not tried	No	
Nav13_02-02-23-001_005_ch1.dat	-1.10321	80	152	16.78	Yes	Yes	not measured
Nav13_02-02-23-002_005_ch1.dat	-0.637817	80	152	14.95	No	Yes	not measured
Nav13_02-02-23-003_004_ch1.dat	-0.552368	80	152	14.04	No	Yes	not measured
Nav13_02-07-23-001_005_ch1.dat	-0.276184	100	152	7.93	No	Yes	not measured
Nav13_02-07-23-002_003_ch1.dat	-0.0518799	100	152	no depolarization	No	No	44.25
Nav13_02-07-23-003_005_ch1.dat	-0.0473022	100	152	not tried	not tried	No	
Nav13_02-07-23-004_003_ch1.dat	-0.418091	100	152	7.32	No	Yes	not enough sodium current
Nav13_02-07-23-006_003_ch1.dat	-0.131226	100	152	not tried	not tried	Yes	
Nav13_02-08-23-002_006_ch1.dat	-0.917053	100	152	10.07	Yes	Yes	21.97
Nav13_02-08-23-001_003_ch1.dat	-0.0198364	100	152	not tried	not tried	No	
HEK_Nav13_05-23-23-003_002_ch1.dat	-0.561523	98	60	11.14	No	Yes	not measured
HEK_Nav13_05-25-23-#2-005_003_ch1.dat	-0.141907	124	60	not tried	not tried	little	
HEK_Nav13_05-25-23-#2-007_002_ch1.dat	-0.0762939	124	60	not tried	not tried	little	
HEK_Nav13_05-25-23-#2-008_002_ch1.dat	-0.177002	124	60	not tried	not tried	little	
HEK_Nav13_05-25-23-#3-010_003_ch1.dat	-0.288391	124	60	8.39	No	Yes	31.01
HEK_Nav13_05-25-23-002_006_ch1.dat	-0.617981	98	60	15.01	No	Yes	not measured
HEK_Nav13_05-25-23-003_004_ch1.dat	-0.128174	124	152	not tried	not tried	little	

Supplementary Table 2: Success of AP generation in the presence of Gd³⁺ and labeling conditions.

Recording file	Gd ³⁺ (μM)	peak current by UV at -70 mV (nA)	laser power (mW)		ΔV (mV) by UV	AP by UV	labeling	ΔV (mV) by Injecting current
			UV	Blue				
Hek_Nav13-05-27-23-#002_10uMGd-013_005_ch1.dat	10	-0.695801	120	60	15	No	Yes	30mV
Hek_Nav13-05-27-23-001_001_ch1.dat	50	-1.9455	124	150	17.73	No	Yes	not measured
Hek_Nav13-05-27-23-003_003_ch1.dat	50	-0.58136	98	12	17	No	Yes	not measured
Hek_Nav13-05-27-23-#002-20uM-011_006_ch1.dat	20	-1.04218	98	60	18.98	No	Yes	36.87
Hek_Nav13-05-27-23-20uM-008_002_ch1.dat	20	-0.398254	124	60	8.18	No	Yes	not measured

Hek_Nav13-05-27-23- #002-20uM- 009_004_ch1.dat	20	-0.198364	124	60	6.81	No	Yes	not measured
Nav13_05-30-23-5uM- #001-002_002_ch1.dat	5	-0.0137329	124	150	not tried	not tried	No	
Nav13_05-30-23-5uM- #001-003_003_ch1.dat	5	-0.294495	124	150	not tried	not tried	Little	
Nav13_05-30-23-5uM- #001-004_005_ch1.dat	5	-0.187683	124	150	not tried	not tried	No	
Nav13_05-30-23-5uM- #002-005_004_ch1.dat	5	-0.0167847	124	150	not tried	not tried	No	
Nav13_05-30-23-5uM- #002-006_003_ch1.dat	5	-0.0534058	124	150	not tried	not tried	No	
Nav13_05-30-23-5uM- #002-009_004_ch1.dat	5	-0.0534058	124	150	not tried	not tried	little	
Nav13_05-30-23-5uM- #003-010_002_ch1.dat	5	-0.125122	98	12	not tried	not tried	little	
Nav13_05-30-23-5uM- #003-012_005_ch1.dat	5	-0.0778198	120	60	not tried	not tried	little	
Nav13_05-30-23-5uM- #003-011_005_ch1.dat	5	-0.439453	120	60	6.21	No	Yes	not measured
Nav13_05-30-23-5uM- #003-013_005_ch1.dat	5	-0.265503	124	150	5.83	No	Yes	35.13
Nav13_05-30-23-5uM- #003-014_002_ch1.dat	5	-0.163269	124	150	3.78	No	Yes	not measured
Nav13_05-30-23-5uM- #003-015_002_ch1.dat	5	-0.0732422	124	150	not tried	not tried	little	32.14
Nav13_05-30-23-5uM- #003-016_004_ch1.dat	5	-0.624084	124	150	12.56	No	Yes	not measured
Nav13_05-30-23-5uM- 017_003_ch2.dat	5	not recorded	124	150	8.45	Yes	Yes	not measured
Hek_13_06-06-23- 001_004_ch1.dat	10	-0.135803	124	150	not tried	not tried	little	
Hek_13_06-06-23- 007_002_ch1.dat	10	-0.349426	124	150	6.4	No	Yes	39.25
Hek_13_06-06-23- 008_002_ch1.dat	10	-0.500488	124	150	9.22	No	Yes	not measured
Hek_13_06-06-23- 003_002_ch1.dat	10	-0.219727	124	150	not tried	not tried	No	
Hek_13_06-06-23- 005_001_ch1.dat	10	-0.0198364	124	150	not tried	not tried	No	
Hek_13_06-07-23- #Dish_2-010uM_GdCl- 005_003_ch1.dat	10	-0.300598	124	150	not tried	not tried	little	
Hek_13_06-07-23- 001_005_ch1.dat	10	-0.489807	124	12	3.78	No	Yes	23.41
Hek_13_06-07-23- 002_002_ch1.dat	10	-3.66211	124	150	not tried	not tried	Yes	
Hek_13_06-07-23- 003_009_ch1.dat	10	-0.668335	124	150	9.19	Yes	Yes	not measured
Hek_13_06-07-23- 0015uM_GdCl_001_ch1.dat	15	-1.4267	124	150	not tried	not tried	Yes	

We considered the labeling with OptoDARg successful when we observed outward optocapacitive currents at -70 mV elicited by the blue laser. This rules out any possibility of thermal effects induced by blue light exposure, since heating would cause $dC/dt > 0$ and this would result in an inward (negative) current (Eq. 3). Indeed, the generation of outward currents at negative holding potentials serves as a hallmark of the photolipid (OptoDARg)-based optocapacitive approach. When the magnitude of the (negative) inward current induced by the UV laser was larger than 200 pA (i.e., the inward current was

less than -200 pA), we considered a success in the labeling, but when a small amount of photolipids was incorporated (magnitude of inward current smaller than 200 pA), we did not try to elicit action potentials because this low amount of current will not generate enough depolarization to elicit an AP.

Confirming the results obtained without photolipid, it can be observed in the updated Supplementary Fig. 4 that when the labeling procedure failed, no capacitive current was generated at -70 mV by UV – both in the presence and absence of Gd^{3+} in the bath. Therefore, we did not try current-clamp experiments as shown in the Supplementary Tables 1 and 2. Eventually, the success rate for AP generation by UV in the absence of Gd^{3+} was 33.3% and in the presence of Gd^{3+} was 13.3%. We believe that the low success rate for the condition without Gd^{3+} is due to the variability in cells and conditions, as outlined above. The rate of success is smaller in the presence of Gd^{3+} due to the fact that Gd^{3+} is blocking Na^+ channels and/or shifting their voltage-dependence to the right (to more positive voltages) by changing the surface potential; this makes it harder for a given depolarization to induce an AP.

When we compare the amount of depolarization induced by UV light-driven photoisomerization of OptoDARg under current-clamp conditions (column “ ΔV (mV) by UV” in the tables), we can observe that it varies from as little as 4–5 mV to up to almost 20 mV. This variability is expected considering the apparent variability in fractional capacitance changes shown in Fig. 3f. Additionally, there are instances where the UV-mediated depolarization was equal to ≈ 7 mV and it was sufficient to elicit an AP, whereas for another cell it was equal to approximately 15 mV and no AP was generated. Therefore, we believe that our low success rates are more due to cell variability than due to the photolipid optocapacitance technique. The success rate of labeling with OptoDARg was approximately the same in the presence (56.7%) or absence of Gd^{3+} (59%). These values are similar to those reported in literature⁶². This similarity of the success rates of labeling is expected since Gd^{3+} was not present during the labeling procedure: it was added after the labeling with OptoDARg was finished.

We modified text in the *Results* section:

page 11: “OptoDARg triggers photo-induced APs

As entailed by Eq. 2, optocapacitive currents modulate the membrane potential. To explore this, we conducted whole-cell recordings in current-clamp mode using HEK293 permanently transfected with $Na_v1.3$. To initiate an AP, only Na^+ currents carried by Na_v channels are necessary⁴⁷, which will depolarize the cell membrane. In our case, we exploited endogenous HEK293 leak channels and leakage currents through the seal to repolarize the cell membrane. Therefore, the HEK293 $Na_v1.3$ can be regarded as an artificial neuron regarding the generation of an AP. In HEK293 $Na_v1.3$, exposure to UV light generated rapid optocapacitive depolarization (Fig. 4a), in agreement with the observation of depolarizing capacitive currents under voltage-clamp (Fig. 3c). Depolarization subsequently triggered the opening of $Na_v1.3$ channels which generated an action potential (Fig. 4a). When there was no OptoDARg incorporated in the cells, there was neither depolarization (Supplementary Fig. 4d) nor optocapacitive current generated upon UV light exposure (Supplementary Fig. 4a,b), which refutes UV light-triggered thermal effects. To the best of our knowledge, this is the first demonstration of direct photolipid-induced depolarization triggering an AP.”

page 11: “The success rate for AP generation by UV in the absence of Gd^{3+} was 33.3% and in the presence of Gd^{3+} was 13.3% (see Supplementary Tables 1 and 2). We believe that the moderate success rate for the condition without Gd^{3+} is due to the variability in cells. The rate of success is smaller in the presence of Gd^{3+} due to the fact that Gd^{3+} might be blocking Na^+ channels and/or shifting their voltage-dependence to the right (to more positive voltages) by changing the surface potential; this makes it

harder for a given depolarization to induce an AP. When we compare the amount of depolarization induced by UV light-driven photoisomerization of OptoDARg under current-clamp conditions (column “ ΔV (mV) by UV” in Supplementary Tables 1 and 2), we can observe that it varies from as little as 4–5 mV to up to almost 20 mV. This variability is expected considering the apparent variability in fractional capacitance changes shown in Fig. 3f. Additionally, there are instances where the UV-mediated depolarization was ≈ 7 mV and it was sufficient to elicit an AP, whereas for another cell it was ≈ 15 mV and no AP was generated. Again, this indicates that low success rates are due to the cells’ condition prior to excitation rather than due to the photolipid technique’s capability to generate depolarization. To elicit an AP in a cell many factors play a role. For example, the amount of sodium channels expressed, the current carried by them and the level of inactivation that they have prior to the depolarizing stimulus. Another factor is related to the input resistance of the cell. In our hands, the cells expressing Nav1.3 required a large depolarization to elicit an AP and sometimes even with a depolarization of 30–35 mV cells failed to fire (Supplementary Fig. 5), whereas in other cases 20 mV was enough to elicit an AP (column “ ΔV (mV) by injecting current” in Supplementary Tables 1 and 2), emphasizing the cellular variability. Therefore, we conclude that our low success rates for evoking APs by OptoDARg-mediated depolarization are due to cell variability rather than due to the photolipid optocapacitance technique.”

Further, we provided more detail in the *Methods* section:

page 18: “Cell labeling

To label the HEK293 cells with the OptoDARg we followed the protocol established by Leinders-Zufall, et al. ⁶⁴. Briefly, prior to labeling, the stock solution containing 60 mM of OptoDARg, dissolved in dimethyl sulfoxide (DMSO - 276855, Sigma-Aldrich), was heated for 5 minutes at 37 °C. The stock solution was diluted with recording/bath solution to the final concentration of 60 μ M. In some cases, to increase the amount and efficiency of labeling, we used 120 μ M OptoDARg (indicated in the text when used). The solution was then vortexed and incubated at 37 °C for 5 minutes. The dish containing the cells was taken from the incubator and washed 1–3 times with the external solution. After washing the dish containing the cells, we added the solution containing OptoDARg and incubated it at 37 °C for 1 hour. After labeling, we washed the dish 1–3 times with the external solution and assembled it in the recording chamber. The stock solution containing the OptoDARg was aliquoted in small volumes (≈ 6 μ l) and stored at –20 °C. The DMSO concentration never exceeds 0.2% in the experiments.

We considered the OptoDARg labeling (incorporation) successful when we observed outward currents at –70 mV elicited by the blue laser. This rules out thermal optocapacitive effects induced by blue light exposure since heating would cause $dC/dt > 0$ and this would result in an inward (negative) current (Eq. 3). Indeed, the generation of outward currents at negative holding potentials was a hallmark of the photolipid (OptoDARg)-based optocapacitive approach (Supplementary Fig. 4c). When the magnitude of the (negative) inward current induced by the UV laser was larger than 200 pA, we considered a success in the labeling. When a small amount of photolipids was incorporated (inward current magnitude of less than 200 pA), we did not try to elicit an AP because not enough depolarization for eliciting an AP would have been generated. Confirming the results obtained without photolipid, it can be observed in Supplementary Fig. 4 that when the incorporation procedure failed, no optocapacitive current was generated at –70 mV by UV – both in the presence and absence of Gd^{3+} in the bath. Therefore, we did not proceed with current-clamp experiments, as detailed in Supplementary Tables 1 and 2. The success rate of labeling with OptoDARg was approximately the same in the presence (56.6%) or absence of Gd^{3+} (59%). These values are similar to those reported in literature ⁶⁴. This similarity is

expected since Gd^{3+} was not present during the incorporation procedure: it was added after the labeling with OptoDARg was finished.”]]

5C) More details about the experimental and biological replicates are needed. Especially in Figure 4 and 5. Best is to provide a statistical quantification for each claim and then give the numbers of replicates in the figure caption.

[[In Fig. 4 we just reported representative traces for our claims – it is intended to represent a summary of our findings. Please see in the tables above that we have tested approximately 52 cells and all the analyses are reported there. For Fig. 5, we used 3 biological replicates. We updated the figure legend to accommodate the Reviewer’s suggestions. The generation of an AP mediated by mechanosensitive currents using the blue laser was not tried consistently, mainly because we were more interested in the direct AP generation by UV light and not in the indirect generation via the opening of mechanosensitive channels which subsequently facilitate depolarizing currents (i.e., AP generation upon UV light-driven optocapacitive depolarization). Nevertheless, in the absence of Gd^{3+} , we observed that around $\approx 90\%$ of the time the blue laser generated an AP through mechanosensitive currents.]]

5D) The authors need to apply a statistical test for the capacitance decay that they determine in figure 3f. Given the spread of the data in figure 3f, I am unsure if the reported trend is statistically significant. Also, we need to know if the 10 cells have been recorded in a single session, or if these are biological replicates from fresh preparations.

[[The spread of the data can be easily explained. First, it comes from different freshly made preparations. The main difference between them is the amount of labeling of cells with OptoDARG. Not all the cells ended up with the same amount of labeling and thus OptoDARG incorporated in the membrane. The absolute change in capacitance depends on photolipid content in the membrane (Eq. 5). Consequently, we cannot reach the same changes in capacitance if the cells do not contain the same m% of photolipids (OptoDARG) in their membrane, even if we use the full laser power (≈ 124 mW). Nevertheless, we ran a paired t-test statistical analysis and at $P \leq 0.01$ the cellular membrane capacitance in the cis and trans photostationary state is significantly different. Please see below a plot. We also updated Fig. 3f with the statistical data and number of replicates.

5E) We need more quantification of the control, i.e. data without the photolipid. At the moment there is a single trace shown. Given many reports that at similar pulsed laser systems, the thermal effects can lead to currents, this is a sensitive issue. Please demonstrate that the AP is not simply triggered by the lasers, but by the combination of lasers and photolipid. Again, this is important as it is a key claim of the paper.

[[Cell experiments: To address the Reviewer’s concerns about the data without photolipid, we included in the Supplementary Fig. 4 (see response to question #5 above) more traces and a complete description of the experiments can be found in the Supplementary Tables 1 and 2. Please note that there were no

photolipids present in these cells, hence laser illumination produced *no optocapacitive currents* (Supplementary Fig. 4). In the absence of optocapacitive currents, the lasers did not elicit APs.

Planar lipid bilayer experiments: We note that neither optocapacitive currents (exemplarily, Fig. 1f) nor changes in capacitance (exemplarily, Supplementary Fig. 3b) were observed upon exposing planar lipid bilayers without OptoDARg to 50 ms pulses of intense blue or UV light. Moreover, thermally-evoked optocapacitive currents would produce membrane thinning, i.e., $dC/dt > 0$. With blue light, we obtain $dC/dt < 0$.]]

6) The explanation for the action potential using blue light as an effect of membrane tension is very interesting. The authors can here include recent work that shows an increase in the area using another azobenzene molecule (Georgiev et al, 2018, *Advanced Science*), although another paper has recently shown an the opposite effect on membrane area (Hoglsperger 2023, *Nat. Comm*).

[[Indeed, the mechanism of area change by OptoDARg photoisomerization agrees more closely with that reported in Georgiev *et al. Adv. Sci.* **5**, 1800432 (2018). Like the water-soluble orthotetrafluoroazobenzene photoswitch (F-azo) Georgiev *et al.* used, photoisomerization of OptoDARg alters surface area due to differences in molecular structure and consequent orientation in the membrane. As we demonstrated in an earlier publication on microaspirated giant unilamellar vesicles¹⁹, blue light triggers area reduction whilst UV light achieves the opposite. Höglspenger *et al. Nature Communications.* **14**, 3760 (2023) show the opposite effect since their photoswitch tends to leave the membrane upon UV light-driven trans-to-cis photoisomerization. Since both references are successful at altering membrane area, we decided to cite them both:

- **ref. 36:** Georgiev *et al. Adv. Sci.* **5**, 1800432 (2018)
- **ref. 38:** Höglspenger *et al. Nature Communications.* **14**, 3760 (2023)

page 8: "Notably, albeit the underlying molecular mechanisms may be distinct – i.e., other photolipids may alter their oligomeric state or partition in/out of the membrane upon switching – changes in in the area of lipid bilayers^{19,36,37} and cellular membranes^{29,38} upon photoisomerization of membrane-targeted azobenzene-containing molecules have been reported by multiple groups."]]

The simple calculation presented suggests that enough energy is provided to switch mechanosensitive channels. However, this calculation is done for a large membrane area. Following the logic, the distance between triggering channels would be in the order of 6µm (the calculation was done for (40µm)² area)...can this be realistically leading to a trigger of tension-mediated AP?

[[We modified the calculation to show that AP triggering does not require large areas. The reasoning is similar to before, yet we now express energy densities (energy per unit area) to emphasize that the calculations do not depend on the absolute size of the object (in terms of membrane area) but on the photolipid content of its membrane.

page 8: "As a back-of-the-envelope calculation shows, the achievable tension, τ , is relevant for mechanosensitive proteins. For example, mechanosensitive channels like MscL open at $\tau \approx 10$ mN/m³⁹. Such τ value translates for membranes with a typical stretching modulus, K , of 250 mN/m into an area increment of 4% ($\alpha = 0.04$), since $\tau = K \times \alpha$. Thus, the energy, E_r , required to stretch a membrane of area A_M is equal to:

$$\frac{E_r}{A_M} = \frac{1}{2} \tau \times \alpha = 0.2 \frac{\text{mN}}{\text{m}}$$

The work performed by isomerization of a single azobenzene switch has been estimated at $E_p=4.5 \times 10^{-20}$ J⁴⁰. Assuming an OptoDARG concentration of 10 mass-% and an average area per lipid of 0.7 nm^2 , we find one photolipid in an area $A_L=7 \text{ nm}^2$. Since the photolipid contains two azobenzene switches we can write:

$$\frac{E_p}{A_L} = \frac{2 \times 4.5 \times 10^{-20} \text{ J}}{7 \text{ nm}^2} = 13 \frac{\text{mN}}{\text{m}}$$

A comparison of the two above equations reveals that:

$$\frac{E_p}{A_L} \gg \frac{E_r}{A_M}$$

Thus, it is safe to conclude that photolipid switching may provide the energy required to open mechanosensitive channels, even if (i) less than 100% of the photolipids change their conformation, (ii) the deformation of other lipids and flattening of membrane undulations requires some extra energy.”]]

Additionally, the expected m% of the photolipid in the cells may be much smaller than in the PLB experiments used for the estimation.

[[Indeed, in contrast to planar lipid bilayers, we observed that the OptoDARG content in the cellular membranes as estimated by capacitance recordings was less homogeneous (Fig. 3f), as can be expected considering the characteristics of both model systems. Yet, in the successful experiments, the OptoDARG content within planar lipid bilayers and HEK cell membranes was comparable, as demonstrated by measuring capacitance changes upon photoisomerization:

page 9: “Subsequently, we sought to gain a rough estimate for the amount of membrane-embedded OptoDARG. We did so by capacitance measurements, measuring the time integral of the transient current for voltage steps (Fig. 3d). A measurement of C in the cis photostationary state, C_{cis} , and the trans photostationary state, C_{trans} , is shown in Fig. 3e. The result of measurements on 10 cells is shown in the upper panel of Fig. 3f. Subsequent calculation of the percentage change in C results in a distribution of values (lower panel in Fig. 3f), frequently around 5%. Conceivably, the incorporation of OptoDARG in cells was less homogenous than in PLBs as can be seen from the spread of percentage change in C in Fig. 3f. Nevertheless, PLBs doped with 10 m% OptoDARG yielded similar changes in capacitance, suggesting that the cellular membranes may also have contained up to 10 m% OptoDARG (Figs. 2 and 3).”]]

Minor comments:

1) Cell viability: The authors should quantify possible cell death due to the application of the photolipid.

[[We have not performed analysis of cell death after application of photolipid. From what we observe, the cells remain healthy even after 1 hour of labeling with OptoDARG and experiments can be performed several hours after that. Additionally, due to the many washes required, it would be very hard to determine whether the cells eventually died or lifted from the dish due to the photolipids or the washing. We believe that the determination of cell viability in the presence of OptoDARG would be beyond the scope of this work which aims at providing a proof of concept.

Still, we would like to comment on cell viability in the presence of OptoDARG. OptoDARG was previously successfully used as a photoswitchable diacylglycerol (DAG) for the regulation of DAG-sensing transient receptor potential channels in HEK293 cells (Lichtenegger *et al. Nat Chem Biol.* **14**, 396–404 (2018)), the

same cell model system used in our study. Unfortunately, no report of cell viability in the presence of OptoDARg exists to date. Yet, for other azobenzene-containing photolipids/fatty acids, such data can be found in literature:

- Chander *et al. Small.* **17**, e2008198 (2021), ref. 44: It was observed that azo-PC, an azobenzene-containing phosphatidylcholine photolipid, when incorporated into lipid vesicles (creating photoactivatable lipid nanoparticles) and subjected to cells “without UV irradiation did not affect cell [Huh7] viability” over 24 h.
- Jiménez-Rojo *et al. bioRxiv* 2022.02.14.480333, ref. 45: In a preprint study, the authors report that upon exposing HeLa cells to 50 μm FAAzo4 for 24 h “the cell viability was not compromised, indicating that AzoPC [see above] was well tolerated in membranes of live cells”. Notably, the cellular tolerance to photolipids is further supported by the same study which reports that HeLa cells can incorporate FAAzo4, a synthetic azobenzene-containing fatty acid, into glycerophospholipids, thereby generating photoswitchable lipids.

page 9: “We observed that cells remained healthy after 1 h of labeling and patch-clamp experiments could be performed for hours after. The azobenzene photoswitch generally combines good photophysical and pharmacokinetic properties with low phototoxicity, e.g., low singlet oxygen generation¹⁴. In cellular systems under experimental conditions, exposing HuH7 cells to lipid vesicles comprising the azobenzene-containing photoswitchable phospholipid azo-PC does not impact cellular viability in the absence of irradiation for 24 h⁴⁴. The cellular tolerance to photolipids is further supported by a recent preprint study which reports that HeLa cells can incorporate FAAzo4, a synthetic azobenzene-containing fatty acid, into glycerophospholipids, thereby generating photoswitchable lipids without compromising cellular viability over the course of 24 h⁴⁵.”

2) The HEK cells are not listed anymore at ATCC. It would be useful to mention this.

[[Thanks. We updated the text.

page 18: “HEK293 cells stably transfected with Nav1.3 channels were purchased from American Type Culture Collection (CRL-3269 (currently discontinued), ATCC, Manassas, VA). The cells, at the time they were acquired, were expressing the Kir2.1 K⁺ channel. However, after 7–8 passages, the cells lost the ability to express Kir2.1. Therefore, all experiments were performed under the condition that no Kir2.1 was present, as it can be observed in the representative Na⁺ current traces shown in Supplementary Fig. 4e.”]]

3) The stably transfected HEK cells also express an additional potassium channel. To which extent can this additional channel interfere with the results reported.

[[The cells, at the time we bought them, were supposed to have the Kir2.1 channel that could be used to set the resting potential of the cells according to the concentration of the K⁺ in the bath and in the pipette. However, after 7–8 passages, the cells lost the Kir2.1 channel expression. This was disclosed to us after we bought the cells and contacted ATCC reporting that we no longer had the Kir2.1. All the experiments were performed under the condition that no Kir2.1 channels were present, as it can be observed in the representative Na⁺ current traces shown in Supplementary Fig. 4. Please see above the text we have updated. Kir2.1 did not interfere since it was not present at the time the experiments were performed.]]

Reviewer #2

The manuscript introduces a method for regulating neuronal activity using photolipids, specifically azobenzene-containing diacylglycerols, without the need for genetic modifications. These photolipids mediate light-triggered cellular de- or hyperpolarization. Upon exposure to UV light, photolipid conformation alters, embedding plasma membranes with increased capacitance and allegedly inducing depolarizing currents, which subsequently trigger voltage-gated sodium channels in cells, leading to action potentials. In contrast, blue light decreases membrane capacitance, allegedly causing hyperpolarization. This approach allows for precise cellular excitation through controlled illumination and self-insertion of photolipids into membranes.

[[We appreciate the positive feedback and the questions raised by the Reviewer. Please see below our comments in blue.]]

While the manuscript offers an insightful exploration into the capabilities of OptoDARg as a light-switch molecule, it would be significantly beneficial to readers if the authors could provide a comparative analysis with previously reported molecules possessing similar mechanisms of action. In particular, the molecule Ziapin2, as detailed in *Nature Nanotechnology* 15, 296–306 (2020), and the light-activated molecular machines described in *Nature Nanotechnology* 18, 1051–1059 (2023), stand out as pertinent comparatives. Drawing parallels and contrasts with these molecules would not only underscore the novelty or advantages of OptoDARg but also offer a comprehensive landscape of the advancements with pros/cons analysis in the field. I would strongly urge the authors to incorporate this comparison to enrich the discourse and contextual relevance of their findings.

[[As suggested we cite both papers, deliberate on Ziapin2 in more detail, and draw parallels and contrast OptoDARg with Ziapin2 and molecular motors.

page 2f: “Recently, neuronal excitation was achieved using the membrane-partitioning azobenzene-containing photoswitchable compound Ziapin2²⁹. The photoisomerization of membrane-embedded trans-Ziapin2 by blue light led to a rapid decrease in membrane capacitance due to a change in Ziapin2’s oligomeric state²⁹. Conceivably, this drop produced a hyperpolarizing capacitive current. Yet, conductances of other origins also contributed as the observed hyperpolarizing current (i) did not linearly depend on voltage and (ii) persisted nearly unaltered for ≈250 ms even though the capacitance change took no longer than ≈20 ms³⁰. Subsequently, delayed depolarization ensued by an elusive mechanism. Being unaware of the channels that might have facilitated the depolarizing current, the authors speculated that Ziapin2 caused a spontaneous fast capacitance increase ($t_{1/2} \approx 0.2$ s). However, Ziapin2’s unprompted return to its cis conformation in DMSO takes orders of magnitude longer ($t_{1/2} = 108$ s)³⁰.”

page 3: “Consequently, photoisomerization of membrane-embedded OptoDARg changes lipid bilayer capacitance, C:

$$C = \varepsilon \times \frac{A}{d_{hc}} \quad (1)$$

where ε denotes absolute permittivity which we assume to be constant. Notably, the difference in the shape of the two monomeric OptoDARg photoisomers drives the changes in d_{hc} . In contrast, the recently reported Ziapin2 decreases d_{hc} due to spontaneous dimer formation and increases d_{hc} upon blue light-driven dimer dissociation²⁹.”

page 14: "We propose a photolipid-based approach to induce direct depolarization and AP generation in excitable cells by light. The approach does not require the presence of intermediary mechanosensitive channels that would translate a light-induced increase in membrane tension into depolarizing currents. Additionally, upon blue light exposure, OptoDARg causes initial hyperpolarization of the membrane potential. We find that in the presence of mechanosensitive channels, the lateral stress generated upon transitioning from cis- to trans-OptoDARg may trigger depolarizing mechanosensitive currents, causing delayed depolarization. The recently developed photoswitch Ziapin2²⁹ induces the same response upon blue light excitation. Yet, only OptoDARg is fundamentally applicable to exert bidirectional direct optocapacitive control over the membrane potential."

page 14: "For comparison, calcium waves in HEK cells were stimulated by exposing molecular motors to 320 Wcm⁻² of 400 nm light for typically 250 ms⁵⁹. That is, the underlying activation of inositol triphosphate signaling pathways used a total dose of 80 Jcm⁻²."]

I would like some clarification regarding the sign of the transmembrane optocapacitive current, I_{cap} , presented in the paper. Specifically, referring to equation (3), it suggests a negative, hyperpolarizing current when V is negative, such as $V = -130$ mV during UV light illumination, especially considering dC/dt is positive during membrane thinning under UV illumination. A negative current, which effectively increases the charge on the capacitor with an increasing capacitance due to membrane thinning, should produce a hyperpolarizing effect. Yet, in Figures 3 and 4, the UV light appears to have a depolarization effect, suggesting a positive I_{cap} under UV illumination. Please clarify.

[[We believe that there is a misunderstanding about the negative current. Indeed, under the conditions mentioned by the Reviewer, the optocapacitive current generated upon UV illumination would be negative. Yet, "by convention, outward membrane currents always are considered positive [...], whereas inward currents are considered negative" (Bertil Hille, *Ion channels of excitable membranes* (Sinauer Associates, Sunderland, Mass., ed. 3, 2001)). But an inward-directed current of sodium or potassium is depolarizing. Hence, negative current leads to a depolarization and not a hyperpolarization of the cellular membrane.]]

A significant point of concern arises from the absence of characterizing local temperature changes during illumination. While the authors assert that the photoswitching molecule presents a distinct mechanism from the traditional photothermal-based optocapacitive mechanisms, the lack of temperature measurements during illumination undermines this argument. Without these crucial measurements, the possibility remains that there may still be photothermal effects (especially at high intensities such as shown in Fig. 1e) contributing to the observed changes, even if unintentional.

[[**Cell experiments:** We share the Reviewer's concern about temperature effects. Yet, measurements of temperature may not be necessary since membrane capacitance is a perfect indicator of temperature changes (Bassetto *et al. Biophys J.* **122**, 661–671 (2023)). Our capacitance recordings indicate that temperature effects are not involved: First, Fig. 5d shows a decrease in capacitance upon blue light illumination because cis-to-trans photoisomerization of OptoDARg by blue light reduces capacitance, while heating would induce an increase (e.g., Shapiro *et al. Nature Communications.* **3**, 736 (2012)). We could observe thermal optocapacitive effects when the blue laser power was larger than 200 mW (see new Supplementary Fig. 4). To avoid these thermal effects, in the experiments conducted for this study the power of the blue laser **NEVER** exceeded 160mW. Second, the capacitance does not change when the UV laser is illuminating the cells in the absence of OptoDARg, as can be judged from the absence of optocapacitive currents (see updated Supplementary Fig. 4). Third, thermal effects in the UV would cause an increased current as long as the illumination lasts (see Shapiro *et al. Nature Communications.* **3**, 736 (2012)), yet photolipid induced currents decayed to baseline during illumination (Figs. 3 and 5)!

To address the Reviewer's concern, we added more control traces in the updated Supplementary Fig. 4 and two tables (Supplementary Tables 1 and 2; see response to Reviewer #1 above) with all the information regarding power, success rate, etc. As it can be seen in Supplementary Fig. 4 and in the tables, there is no evidence that supports a thermal effect. Further, we modified the text in the *Results* section:

page 9: "Photolipid-evoked hyper- and depolarizing currents in cells

As in PLBs, we found that exposing the cells (Fig. 3b) to millisecond UV and blue light pulses (445 nm) generated hyper- and depolarizing currents, respectively (Fig. 3c). In accordance with Eq. 7 and in line with our PLB recordings (e.g., Fig. 1e), the optocapacitive currents in Fig. 3c decay exponentially. This is in stark contrast to a thermal optocapacitive mechanism in which light exposure causes an increased current as long as the illumination lasts⁸. Further, no optocapacitive current was observed when cells were not labeled with OptoDARg (Supplementary Fig. 4a,b), ruling out the possibility that the currents shown in Fig. 3c were thermally evoked. The conclusion is based on the observations that membrane capacitance is a perfect indicator of temperature changes^{8,46} and that C did not change upon laser illumination in the absence of the photolipid. It is important to note that when the blue laser power exceeded 200 mW, it was possible to observe thermal optocapacitive effects (Supplementary Fig. 4c). Therefore, we never exceeded 160 mW when conducting experiments using the blue laser."

page 13: "Fig. 5d highlights the rapid ms-timescale changes in capacitance of OptoDARg-containing cellular membranes upon UV and blue light illumination that are critical to the optocapacitive method^{10,13}. Coincidentally, Fig. 5d refutes underlying thermal effects⁸: (i) heating by blue light would have induced an increase in C, not the demonstrated decrease, and (ii) prolonged exposure to UV and blue laser light did not lead to the prolonged changes in C that characterize photothermal events⁸."

Planar lipid bilayer experiments: In the absence of OptoDARg, we do not observe changes in capacitance upon intense UV or blue light exposure (Supplementary Fig. 3b) – intensity and duration of exposure are as in Fig. 1e. Likewise, in Fig. 1f no optocapacitive currents are observed in the absence of OptoDARg. Hence, we can exclude a photothermal optocapacitive effect due to heating of the membrane by our laser illumination in the absence of OptoDARg. Further, if OptoDARg constituted an effective photothermal converter – which appears unlikely as it is a switch and not performing work continuously as a molecular motor might do – the result of intense blue light exposure in e.g. Fig. 1e would be membrane thinning (i.e., $dC/dt > 0$) which would generate optocapacitive currents in the opposite direction (negative currents at negative holding potential, compare also Supplementary Fig 4c). The effect observed in Fig. 1e (positive capacitive currents at negative holding potential) cannot be induced thermally. If unintentional thermal effects were present, they could only diminish the current amplitudes shown in Fig. 1e. We have modified the text to address the Reviewer's concern.

page 4: "Since photoisomerization from cis- to trans-OptoDARg by blue light reduces C, this result is in line with Eq. 3. Further, this observation rules out a photothermal optocapacitive mechanism induced by blue light because heating thins the membrane, i.e., mandates $dC/dt > 0$ which would generate oppositely-directed I_{cap} ^{8,10,11}.""]

In examining Figures 1c and 1d, I observed distinct disparities in the timescales of the optocapacitive currents between the cis-to-trans and trans-to-cis transformations. Specifically, the trans-to-cis transformation depicted in Figure 1d seems to manifest at a considerably slower pace (almost 10 ms)

compared to the cis-to-trans transformation in Figure 1c (1-2 ms). Could the authors elucidate the underlying reasons for this pronounced difference?

[[The product of irradiance-normalized photoisomerization rate and irradiance determines capacitive current decay (Eq. 7). Since the irradiance-normalized rates of azobenzene photoisomerization from cis to trans with blue light and vice versa with UV light are similar (Arya *et al. J Chem Phys* **152**, 24904 (2020)), the lower UV irradiance is responsible for the disparity in the rate of optocapacitive current decay. The dependence of optocapacitive current decay rate on irradiance can be appreciated from Supplementary Fig. 1 – the higher the irradiance, the faster the current decay. The lower irradiance with UV is due to the distribution of the UV laser light over a larger area. We added text to the caption of Fig. 1 to address the apparent discrepancy and explain its origin.

page 5: "In comparison to **c**, maximum current amplitude and rate of current decay are apparently reduced; as predicted by Eq. 7, this is a consequence of the lower irradiance of UV relative to blue light (see Methods)."]]

This point is worth discussing especially because the authors claim that the millisecond level current under UV light (which is actually close to 10 ms) has a depolarizing effect to trigger action potentials.

[[**Cell experiments:** The UV light irradiance in the cell experiments was higher (compare Fig. 1d with Fig. 3c) and chosen empirically to suffice at generating sufficient depolarization to elicit APs. We generated optocapacitive depolarization and APs using $\approx 5\text{--}6$ ms pulses of UV light, limited by the deployed mechanical shutter. Yet, as Fig 5d shows, the change in capacitance by UV light already reaches a plateau after $\approx 1.5\text{--}2$ ms. As mentioned above, we were limited by the mechanical shutter which did not allow us to use shorter excitation times. Beyond the proof of concept demonstrated here, the photolipid-based optocapacitive approach will be improved and optimized in the future. The updated Methods section (especially, "Optical stimulation setup") now more clearly outlines differences in the laser irradiances used between planar lipid bilayer and cell experiments.]]

Reviewer #3

The work reports on a new method for modulating the electrical behavior in excitable cells by using light and photolipids. The work is clear, well written and the topic is of high interest. The reported measurements are of high quality and the analysis of the results look comprehensive. In light of the above I recommend publication after minor revisions.

[[We thank for the positive feedback and the questions raised by the Reviewer. Please see below our comments in blue.]]

- The authors suppose that permittivity of the membrane is constant upon light exposure, mechanical stress and rearrangement (see equation 1 and the following analysis through the whole text). Hence, they base the whole data analyses on the variation of membrane capacitance due to membrane thickness and area. Indeed, this is not proved. Molecular polarizability may change upon rearrangement of lipid layers and, for example, water or salt ions solvation may change and/or ions may penetrate in between lipid molecules. This may significantly alter the polarizability and then the permittivity. I agree that this is expected to be a minor contribution however, as far as I know, this has never been investigated and then proven. Hence, I recommend to the authors to clearly state that their analyses are based on this assumption unless they can show the opposite or cite previous literature.

[[Correct, for simplicity we assume that the permittivity of PLBs and plasma membranes remains constant upon photoisomerization. We added an explicit statement of this assumption to the manuscript.

page 3: "Consequently, photoisomerization of membrane-embedded OptoDARg changes lipid bilayer capacitance, C:

$$C = \epsilon \times \frac{A}{d_{hc}} \quad (1)$$

where ϵ denotes absolute permittivity which we assume to be constant."]]

- It's worth noticing that the light power used to switch the isomer conformation is in the order of 100 Watt per cm² in the UV region (in the visible is even higher). This is a very high value that may compromise practical applications. I think that the approach can be improved and optimized in the future and the aim of the present work is to prove the concept and its functioning. However, the author should better highlight this crucial point in the text and, in particular in the conclusion.

[[

page 14f: "Whilst light power and photolipid concentration allow for adjustable modulation of the membrane potential, it is worth noticing that we used high irradiances of up to 4 kWcm⁻² (UV) and 15 kWcm⁻² (blue) to evoke APs. Yet, trigger pulses on the order of only 1 ms were sufficient to elicit APs (Fig. 4). Consequently, with only 4–15 Jcm⁻², the doses – irradiance times exposure duration – used were relatively low, which should mitigate unwanted side effects. For comparison, calcium waves in HEK cells were stimulated by exposing molecular motors to 320 Wcm⁻² of 400 nm light for typically 250 ms⁵⁹. That is, the underlying activation of inositol triphosphate signaling pathways used a total dose of 80 Jcm⁻². Further, triggering APs in neurons in the presence of gold nanoparticles by a photothermal optocapacitive approach required 31 kWcm⁻² × 1 ms = 31 Jcm⁻² of 532 nm light⁹. That is, the use of photolipids appears advantageous as the photoeffects are induced at a fraction of the energy injected by comparable methods. By delivering light in an even shorter interval in the μ s-time range¹⁰, the dose to elicit an AP by the optocapacitive approach can be reduced even further. Beyond the proof of concept

demonstrated here, this emphasizes that the photolipid-based approach can be improved in the future, e.g., by more efficient photolipids.”]]

- Still regarding the discussion and the conclusion I would pay attention to state that the approach “it is easy to handle since externally added photolipids spontaneously insert into both leaflets of excitable cells”. I agree, however, the lipid membrane is the result of 1 billion year of evolution. It does play a fundamental and very complex role in cell biology. Its modification with no side effects it is not trivial to achieve or to predict. Hence, I would be more cautious, and I would clearly say further studies are necessary to ensure that there are no side effects due to membrane modifications.

[[We added a note of caution, appreciating the complexity of the biological membrane and its physiological responses to the Conclusion.

page 15: “The photolipid-based optocapacitive approach may be readily implemented considering that externally added OptoDARG spontaneously inserts into both leaflets of excitable cells, providing an elegant alternative method to genetic manipulations. Yet, further studies are required to understand the complexity of physiological effects evoked by photolipid photoisomerization in complex biological membranes. Effectively, our observation that photolipid photoisomerization can trigger mechanosensitive channels constitutes one “side effect” of our photolipid-based optocapacitive approach.”]]

Reviewer #1 (Remarks to the Author):

I would like to thank the authors for the extensive description of the changes made, and the additional explanations provided in the manuscript. All my concerns have been addressed in detail, and from my point of view, the manuscript is now much clearer and more convincing. Therefore, I strongly support publication of the manuscript in its current form in Nature Communications.

Reviewer #2 (Remarks to the Author):

The authors have addressed most of my comments and questions with satisfaction. However, I still have some remaining questions that are unaddressed:

1) When discussing the thermal effect of their approach, the authors cited Fig. 5d, which shows a decrease in capacitance upon blue light illumination. They argued that heating would induce an increase in capacitance (e.g., Shapiro et al. Nature Communications. 3, 736 (2012)), thus ruling out the possibility of heating during blue illumination. I agree with this argument. However, Fig. 5d also shows an increase in capacitance under UV illumination. Since this change is in the same direction as one would expect for membrane temperature increase, one may wonder if this could be caused by heating. Can the authors comment on this observation?

2) I don't agree with the authors that "thermal effects in the UV would cause an increased current as long as the illumination lasts". A lasting illumination with fixed power density will eventually reach thermal equilibrium because cooling becomes more effective as local temperature increases. Therefore, with lasting illumination one should expect a decay in induced current too.

3) The authors' control experiments without OptoDARg do not represent a fair control since OptoDARg has absorption in the UV-blue spectrum, thus leading to greater thermal effect than membranes without.

Reviewer #3 (Remarks to the Author):

The authors addressed the points I raised and I think the manuscript can now be considered for publication in this journal.

I would have appreciated a more clear/extended statement regarding the assumption of constant permittivity.

In my opinion it is a point often underestimated from the community.

However, in this work, it is not a critical point.

Below, the reviewers' comments are reproduced with our replies interspersed in **[[brackets in blue type]]**. In **highlight** is the text added to the manuscript. Quotations from the manuscript are in "quotation marks".

Reviewer #1

I would like to thank the authors for the extensive description of the changes made, and the additional explanations provided in the manuscript. All my concerns have been addressed in detail, and from my point of view, the manuscript is now much clearer and more convincing. Therefore, I strongly support publication of the manuscript in its current form in Nature Communications.

[[We thank the Reviewer once again for her/his extensive comments whose consideration has improved the manuscript and thank her/him for supporting publication of the revised manuscript.]]

Reviewer #2

The authors have addressed most of my comments and questions with satisfaction. However, I still have some remaining questions that are unaddressed:

1) When discussing the thermal effect of their approach, the authors cited Fig. 5d, which shows a decrease in capacitance upon blue light illumination. They argued that heating would induce an increase in capacitance (e.g., Shapiro et al. Nature Communications. 3, 736 (2012)), thus ruling out the possibility of heating during blue illumination. I agree with this argument. However, Fig. 5d also shows an increase in capacitance under UV illumination. Since this change is in the same direction as one would expect for membrane temperature increase, one may wonder if this could be caused by heating. Can the authors comment on this observation?

[[If thermal effects were present in our experiments, we should have observed asymmetric capacitance changes with respect to blue and UV light illumination. In other words, if UV light had exerted a thermal effect, the capacitance should have increased by a combination of two effects: *azobenzene switching* plus *thermally-induced thinning*. Blue illumination causes a decrease in capacitance by switching back the azobenzene moiety. However, it does not cause a *thermally-induced thickening* of the membrane, i.e. it does not cool the membrane. In consequence, if blue and/or UV light had caused heating of OptoDARg-containing membranes, the change in capacitance should have been asymmetric, with UV light inducing a greater change in capacitance than blue light. However, as Fig. 5d clearly demonstrates, we do not observe any such asymmetry. In Fig. 5d, the absolute values of the capacitance changes upon UV and blue light exposure at equilibrium (i.e., after ≈ 5 ms of exposure) are identical within $\approx 10\%$. We therefore conclude that thermal effects, if present, are marginal and do not add significantly to the structural effects induced by photolipid transitions.

We have added a respective note to the manuscript:

page 13: "Coincidentally, Fig. 5d refutes underlying thermal effects ⁸: (i) heating by blue light would have induced an increase in C, not the demonstrated decrease, and (ii) prolonged exposure to UV and blue laser light did not lead to the prolonged changes in C that characterize photothermal events ⁸. Further, the very symmetry of the magnitude of the changes in C with UV and blue light exposure in Fig. 5d precludes a thermal effect. This is because heating with UV and/or blue light would always positively add to the change in C that is due to photolipid switching, thus increasing the increment in C with UV and reducing the decrement in C with blue light exposure; this we do not observe."]]

2) I don't agree with the authors that "thermal effects in the UV would cause an increased current as long as the illumination lasts". A lasting illumination with fixed power density will eventually reach thermal equilibrium because cooling becomes more effective as local temperature increases. Therefore, with lasting illumination one should expect a decay in induced current too.

[[The sentence quoted above refers to the cell illumination times used in the manuscript, which do not exceed 10 ms. At the power used, thermal equilibrium is not reached in this time. Fig. 1c in Shapiro *et al. Nature Communications*. **3**, 736 (2012) can be used as evidence. The cited figure displays thermally induced optocapacitive currents that persist over the entire illumination period of 10 ms. In contrast, the photoswitching of OptoDARg takes only a fraction of the illumination time, i.e., the optocapacitive current decays to zero within less than 3 ms (compare e.g. Figs. 3 and 5).

To accommodate the concern of the reviewer we adapted the manuscript as follows:

page 9: "This is in stark contrast to a thermal optocapacitive mechanism, where exposure to light causes an increase in current until thermal equilibrium is reached, which typically requires much longer than the millisecond pulses used here⁸."]]

3) The authors' control experiments without OptoDARg do not represent a fair control since OptoDARg has absorption in the UV-blue spectrum, thus leading to greater thermal effect than membranes without.

[[The control measurements without OptoDARg are absolutely necessary as other light absorbing substances might be present in the cells. Therefore, we cannot remove the results of these control experiments from our manuscript.

Whilst these control measurements do not exclude thermal effects resulting from the interaction of azobenzene and light, the symmetry of the observed light-evoked capacitance changes does. This symmetry strongly argues against a sizeable contribution of thermal effects upon UV and/or blue light exposure – as elaborated upon in the answer to question #1.

To accommodate the concern of the reviewer we adapted the manuscript as follows:

page 9: "Further, no optocapacitive current was observed when cells were not labeled with OptoDARg (Supplementary Fig. 4a,b), indicating that the currents shown in Fig. 3c were not thermally evoked. The conclusion is based on the observations that membrane capacitance is a perfect indicator of temperature changes^{8,46} and C did not change upon laser illumination in the absence of the photolipid. It is important to note that when the blue laser power exceeded 200 mW, it was possible to observe thermal optocapacitive effects (Supplementary Fig. 4c). Therefore, we never exceeded 160 mW when conducting experiments using the blue laser. Whilst these control experiments do not strictly rule out the possibility that the interaction of light and OptoDARg leads to heating, the symmetry of the observed light-evoked capacitance changes upon UV and blue light illumination does (for details see further below)."]]

Reviewer #3

The authors addressed the points I raised and I think the manuscript can now be considered for publication in this journal. I would have appreciated a more clear/extended statement regarding the assumption of constant permittivity. In my opinion it is point often underestimated from the community. However, in this work, it is not a critical point.

[[We thank the Reviewer for reviewing our manuscript and her/his positive response regarding the changes made.]]